# An elastic element in the protocadherin-15 tip link of the inner ear

Raul Araya-Secchi[1], Brandon L. Neel[1] & Marcos Sotomayor[1]

Tip link filaments convey force and gate inner-ear hair-cell transduction channels to mediate perception of sound and head movements. Cadherin-23 and protocadherin-15 form tip links through a calcium-dependent interaction of their extracellular domains made of multiple extracellular cadherin (EC) repeats. These repeats are structurally similar, but not identical in sequence, often featuring linkers with conserved calcium-binding sites that confer mechanical strength to them. Here we present the X-ray crystal structures of human protocadherin-15 EC8–EC10 and mouse EC9–EC10, which show an EC8–9 canonical-like calcium-binding linker, and an EC9–10 calcium-free linker that alters the linear arrangement of EC repeats. Molecular dynamics simulations and small-angle X-ray scattering experiments support this non-linear conformation. Simulations also suggest that unbending of EC9–10 confers some elasticity to otherwise rigid tip links. The new structure provides a first view of protocadherin-15's non-canonical EC linkers and suggests how they may function in inner-ear mechanotransduction, with implications for other cadherins.

[1] Department of Chemistry and Biochemistry, The Ohio State University, 484 W 12th Avenue, Columbus, Ohio 43210, USA. Correspondence and requests for materials should be addressed to M.S. (email: sotomayor.8@osu.edu).

In the vertebrate inner ear, sound and head movements are transformed into electrical signals by specialized mechano-receptors called hair cells[1–3]. These cells feature a bundle of actin-filled projections (stereocilia) arranged in a staircase formation that get deflected upon mechanical stimulation (Fig. 1a)[4]. The tip of each stereocilium is linked to its tallest neighbor by a 'tip link' filament essential for hair-cell mechano-transduction (Fig. 1b)[5–9]. Tip links convey force and gate inner-ear transduction channels to initiate sensory perception[10].

Biophysical studies have shown that gating of hair-cell channels is directly mediated by a soft 'gating spring', which could be in series with the tip link, or be the tip link itself[11–14]. After channel opening by the gating spring, transduction currents decrease in two phases through fast and slow adaptation[3,15–18]. Myosin motors are thought to mediate slow adaptation[19], while the mechanism underlying fast adaptation, including a possible calcium-dependent release element, is not completely understood[2].

Identification of the molecular components of the hair-cell transduction machinery has been challenging[20]. The exact molecular composition of the hair-cell transduction channels is still controversial, but a consensus on the molecular components of tip links has recently emerged. Immature tip links are likely formed by homotetrameric protocadherin-15 (PCDH15) proteins[21], while mature tip links are made of cadherin-23 (CDH23) and PCDH15 heterotetramers (Fig. 1c)[22–27].

Cadherins form a large superfamily of proteins that include the classical cadherins mediating calcium-dependent cell–cell adhesion[28,29]. Most members of the superfamily have an N-terminal extracellular domain made of five to six extracellular cadherin (EC) repeats, followed by a single-pass transmembrane helix and a C-terminal cytoplasmic domain. CDH23 and PCDH15 are unusual members of the family that feature 27 and 11 EC repeats, respectively (Fig. 1c)[30,31]. The ultrastructure of tip links[6] and molecular dynamics (MD) simulations of CDH23 EC repeats with canonical calcium-binding sites[32] suggest that tip links are stiff and may not form the long-sought gating spring, yet the elasticity of the entire complex and of non-canonical EC repeats is unknown. Recent sequence analyses and structures of cadherins have revealed unusual calcium-free inter-repeat linkers in some members of the superfamily[33]. Bound calcium ions have been shown to provide structural rigidity to cadherins[34–36], so the presence of unusual sites may confer flexibility and perhaps affect the tertiary and

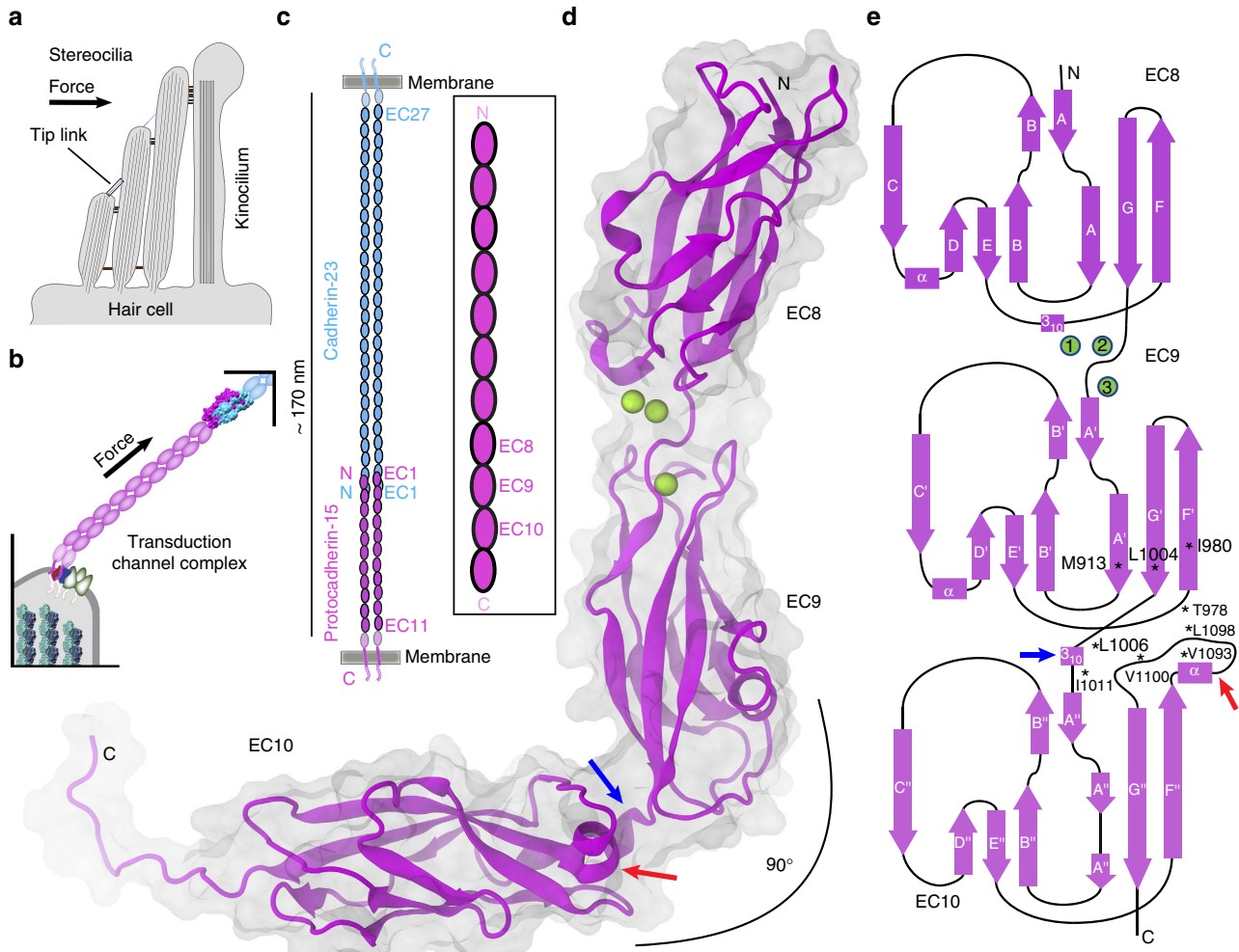

**Figure 1 | Hair-cell mechanotransduction and structure of PCDH15.** (**a**) Schematic representation of a cochlear hair-cell stereocilia bundle highlighting the location of the tip link. (**b**) Mechanotransduction apparatus. PCDH15 directly conveys force to transduction channels. (**c**) The tip link is formed by the tip-to-tip interaction between CDH23 and PCDH15 parallel dimers[25]. Inset shows the location of the repeats studied here. (**d**) Ribbon diagram of PCDH15 EC8–10. Calcium ions in the EC8–9 linker are shown as green spheres. The calcium-free EC9–10 linker is bent. (**e**) Topology diagram of PCDH15 EC8–10. A typical cadherin fold with seven β strands (labeled A to G) is observed for all EC repeats. The structure shows a novel EC9–10 $3_{10}$ helix (blue arrow) at the EC9–10 linker and an atypical EC10 FG-α loop (red arrow). Residues that form the EC9–10 interface are highlighted with an asterisk (*).

quaternary arrangement of the cadherins that harbor them. Analysis of the PCDH15 sequence shows unusual calcium-binding sites in some of its inter-repeat linkers. Here we present the X-ray crystal structures of repeats EC8–10 and EC9–10 refined at 2.8 and 3.35 Å resolution, respectively, which show an EC9–10 calcium-free linker that alters the linear arrangement of PCDH15's EC repeats (Fig. 1d). We suggest that the unusual features of these PCDH15 repeats affect the elastic response of the tip link and are relevant for its function in sensory perception.

## Results

**Overall structure of PCDH15 EC8–10 repeats**. To understand the role of non-canonical tip-link EC repeats, the human proto-cadherin-15 EC8–10 (PCDH15 EC8–10) and the mouse EC9–10 (mmPcdh15 EC9–10) fragments were refolded from inclusion bodies produced in *Escherichia coli* and used for crystallization and structural determination (see Methods). The solved structure of the human PCDH15 EC8–10 (2.8 Å; Table 1; Supplementary Fig. 1) contained 333 residues starting from Ser 795 (EC8) to Glu

1,125 (numbering corresponds to the processed protein) followed by two histidine residues that are part of the purification tag. The structure for mouse Pcdh15 EC9–10 (3.35 Å, Table 1) contained 228 residues starting from Met 898 to Glu 1,125 (in the most complete protomer of the four found in the asymmetric unit). Since both human and mouse structures show similar features, we will focus on the human PCDH15 EC8–10 structure unless otherwise stated. Overall, each of the three EC repeats shows an architecture similar to that of classical cadherins, with the typical Greek-key motif comprised of seven β strands forming a β sandwich fold (Fig. 1d,e). In addition, repeats EC8 and EC9 feature an α-helix between β strands C and D, also observed in some of the CDH23 and PCDH15 EC1–2 repeats[27,32,37]. Even though there are structural similarities with other members of the superfamily, several features make the PCDH15 EC8–10 fragment unique.

The first unique structural feature of PCDH15 EC8–10 is a canonical-like calcium-binding site at the EC8–9 linker with a novel torsion between these EC repeats. In addition, the EC9–10 linker is calcium-free and bent (90°; Fig. 1d), with an EC9–10 $3_{10}$ helix and an EC10 FG-α loop that define the EC9–10 interface. Each of these structural features is discussed below.

**PCDH15 EC8–9 linker shows a canonical-like $Ca^{2+}$-binding site**. The PCH15 EC8–9 sequence contains most of the canonical motifs involved in calcium binding (Supplementary Fig. 2), which typically define three calcium-binding sites, labeled 1 to 3, with several conserved, negatively-charged residues coordinating calcium ions (Fig. 2a–c; $XEX^{BASE}$ and DYE from the first EC repeat, DXNDN at the linker, and DXD and $XDX^{TOP}$ from the second EC repeat). As expected, the structure shows three bound calcium ions. However, a comparison with PCDH15 EC1–2 (Fig. 2b,c) and other cadherins reveals that the distance between calcium ions at positions 2 and 3 is larger in PCDH15 EC8–9 (8.48 Å versus an average of 6.84 Å for 41 canonical linkers; Fig. 2b,c,f; Supplementary Table 1). The most similar arrangement of calcium ions is observed for those found in the C-cadherin EC4–5 linker, with a distance of 7.93 Å, but most, if not all other linkers show a canonical arrangement. In PCDH15 EC8–9, the side chain of Asp 933 in the DX**D** motif, which usually bridges calcium ions at sites 2 and 3, only coordinates calcium at site 3 in a 'canonical-like' architecture. In addition, the overall orientation of EC8 with respect to EC9 appears contorted and bent when compared with that of the CDH23 and PCDH15 EC1–2 repeats and other cadherin structures (Fig. 2e; Supplementary Fig. 3). The canonical DXNDN linker motif is DMNDY (897–901) in PCDH15 EC8–9, with the carbonyl oxygen of Tyr 901 coordinating calcium. It is likely that the bulky and conserved side chain of Tyr 901 and two proline residues after the DXD motif in EC9 (934–935; also conserved, Supplementary Fig. 4) cause the bending and atypical separation observed between ions at sites 2 and 3.

To determine the stability of the canonical-like arrangement of calcium ions, we performed two equilibrium MD simulations of PCDH15 EC8–10 in solution (Table 2; Supplementary Table 2). The unusually large distance between calcium ions at sites 2 and 3 in linker EC8–9 was fairly constant (8.1 ± 0.2 Å) throughout these trajectories (>400 ns; S1b; 500 ns; S1c; Fig. 2f), suggesting that the observed conformation is stable in solution. The orientation of EC9 with respect to EC8 shows some variability, but is maintained over time (Fig. 2d,e; Supplementary Fig. 5a). Overall, our PCDH15 EC8–10 structure and simulations suggest that the EC8–9 linker is distinct from canonical cadherin linkers.

**PCDH15 EC9–10 linker is bent and lacks $Ca^{2+}$-binding sites**. Sequence analysis indicates the absence of canonical calcium-binding motifs in the PCDH15 EC9–10 linker across different

| | HsPCDH15 EC8-10 | MmPcdh15 EC9-10* |
|---|---|---|
| *Data collection* | | |
| Space group | P 6₃ 2 2 | P 3₁ |
| Unit cell parameters | | |
| $a, b, c$ (Å) | 157.93, 157.93, 142.90 | 143.54, 143.54, 95.60 |
| $\alpha, \beta, \gamma$ (°) | 90, 90, 120 | 90, 90, 120 |
| Molecules per asymmetric unit | 1 | 4 |
| Beam source | APS-24-ID-C | APS-24-ID-E |
| Wavelength (Å) | 0.9792 | 0.9792 |
| Resolution limit (Å) | 2.803 | 3.35 |
| Unique reflections | 26,222 | 31,590 |
| Completeness (%) | 99.4 (99.7) | 100 (100) |
| Redundancy | 7.8 (8.1) | 5.9 (5.8) |
| $I/\sigma(I)$ | 3.77 (4.09) | 6.96 (2.03) |
| $R_{merge}$ | 0.077 (0.68) | 0.256 (0.93) |
| $R_{meas}$ | 0.083 (0.73) | 0.281 (1.02) |
| $R_{pim}$ | 0.029 (0.25) | 0.116 (0.42) |
| $CC_{1/2}$ | 0.977 (0.856) | 0.913 (0.759) |
| $CC^*$ | 0.994 (0.960) | 0.975 (0.929) |
| | | |
| *Refinement* | | |
| Resolution range (Å) | 49.40-2.80 (2.87-2.80) | 46.98-3.35 (3.44-3.35) |
| $R_{work}$ (%) | 17.1 (35.6) | 15.94 (21.5) |
| $R_{free}$ (%) | 19.9 (38.7) | 19.59 (23.9) |
| Residues (atoms) | 333 (2,664) | 900 (7,023) |
| Water molecules | 69 | 25 |
| r.m.s. deviations | | |
| Bond lengths (Å) | 0.011 | 0.010 |
| Bond angles (°) | 1.247 | 1.078 |
| *B*-factor average | | |
| Protein | 72.6 | 78.4 |
| Ligand/ion | 89.79 | 73.63 |
| Water | 60.33 | 34.15 |
| | | |
| *Ramachandran Plot Region (PROCHECK)* | | |
| Most favoured (%) | 90.2 | 85.5 |
| Additionally allowed (%) | 9.4 | 14 |
| Generously allowed (%) | 0.3 | 0.5 |
| Disallowed (%) | 0.0 | 0.0 |
| | | |
| PDB ID code | 4XHZ | 5KJ4 |

*Tertatohedrally twinned. Twin operator (twinning fraction): h, k, l (0.258); −k, −h, −l (0.243); −h, −k, l (0.245); k, h, −l (0.254).

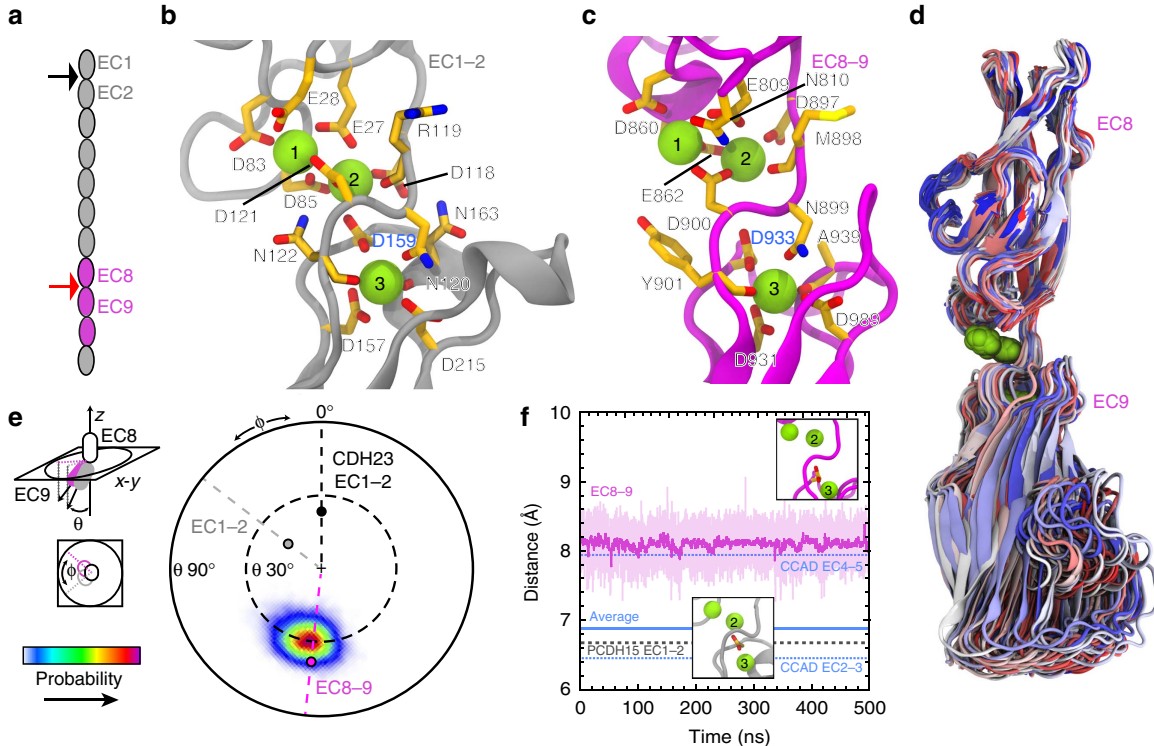

**Figure 2 | Structure and simulated dynamics of PCDH15 EC8–9.** (**a**) Schematics of PCDH15's extracellular domain and the location of linkers EC1–2 (black arrow) and EC8–9 (red arrow). (**b,c**) Detail of calcium-binding sites at the PCDH15 EC1–2 and EC8–9 linkers, respectively. Protein backbone shown as gray (EC1–2) and magenta (EC8–9) ribbon. Relevant residues are shown in stick representation. (**d**) Superposition of PCDH15 EC8–9 conformations taken every 5 ns from an equilibrium MD simulation (S1c; Table 2) using EC8 as a reference. Colour indicates time (blue–white–red). (**e**) To quantify the conformational freedom of EC9 with respect to EC8 the principal axis of EC8 was aligned to the z axis, and the projection in the x–y plane of the principal axis of EC9 was plotted as a probability map for data taken every 20 ps of simulation S1c. The initial orientations for EC8–9, CDH23 EC1–2 (2WHV; $\phi \equiv 0°$), and PCDH15 EC1–2 (4APX) are shown as magenta, black and gray circles, respectively. (**f**) Distance between calcium ions at sites 2 and 3 measured every 20 ps during simulation S1c (pink). A 1 ns running average is shown in magenta; the average for the same distance measured for 41 cadherin linkers (Supplementary Table 1) is shown in blue (dashed lines show maximum and minimum). The same distance measured for PCDH15 EC1–2 is shown as a gray dotted line.

species. The XEX$^{BASE}$, DYE and DXNDN motifs in EC9 are replaced by XAX, NEE and HPG-E, while the DXD and XDX$^{TOP}$ motifs in EC10 are replaced by AIN and DSL (Supplementary Figs 2 and 4). Consistently, the PCDH15 EC9–10 region shows a calcium-free linker that is bent (∼90°) and breaks the linear arrangement of EC repeats. The bending at EC9–10 is similar to that observed in a structure of *Drosophila* N-cadherin (DNcad), which also has a calcium-free linker (PDB codes 3UBH, 3UBG)[33], and to the conformation adopted by T-cadherin (Tcad/CDH13) in calcium-free conditions (3KR5)[38]. However, the PCDH15 EC9–10 linker is L-shaped, while DNcad is V-shaped, and Tcad without calcium is U-shaped (Supplementary Fig. 6a–d). In addition, the PCDH15 EC9–10 linker has structural features not observed in DNcad, Tcad or any other cadherin structures[39–41]. A unique EC9–10 $3_{10}$ helix is located in the middle of the EC9–10 linker (His 1,007–Ile 1,011; blue arrow in Fig. 1d,e), and an atypical EC10 FG-α loop (Leu 1,091–Asn 1,105; red arrow in Fig. 1d,e) form an EC9–10 interface that stabilizes the observed bent conformation of the linker.

The PCDH15 EC9–10 interface can be divided into three components (Fig. 3a,b). The first one is the EC9–10 linker formed by residues Val 1,005 to Arg 1,013, which includes the EC9–10 $3_{10}$ helix. The second corresponds to a mostly hydrophobic core formed by residues Met 913, Thr 978, Ile 980 and Leu 1,004 in EC9 interacting with residues Val 1,093, Leu 1,098 and Val 1,100 in the EC10 FG-α loop. Residues Leu 1,006 and Ile 1,011, which are part of the EC9–10 linker, also participate in hydrophobic

core interactions. The third component of the interface consists of two supporting loops (Val 914–Asp 917 in EC9 and Ala 1,040–Ser 1,046 in EC10) that may provide stability to the linker but do not form part of the interface between EC9 and EC10. Conservation analysis of the interface reveals that most of the hydrophobic residues are highly conserved, while some of the polar and charged residues present in the EC9–10 linker and in the EC10 FG-α loop show more variability (Fig. 3c). The buried surface area of the EC9–10 interface is 389 Å$^2$. A structure of mouse Pcdh15 EC9–10 (without EC8), crystalized in a different condition and space group (Table 1; Supplementary Fig. 7), shows the same unique features. This suggests that the EC9–10 interface is robust and that the observed features are not a crystallographic artifact (Supplementary Discussion).

**PCDH15 EC9–10 bent conformation is stable *in silico*.** The crystallographic, bent conformation of PCDH15 EC9–10 might not be stable in solution and may represent one of many possible arrangements for these repeats. To test the flexibility of the EC9–10 linker we performed equilibrium and steered molecular dynamics (SMD) simulations of the PCDH15 EC8–10 structure in solution.

Two equilibrium MD simulations lasting 400 ns (S1b) and 500 ns (S1c) showed a rather stable bent structure and only local deformation of the EC10 supporting loop (residues 1,040–1,046; Fig. 4a; Supplementary Figs 8a–d and 9a–d). The orientation of

**Table 2 | Overview of MD simulations.**

| Label | # | PDB | $t_{sim}$ (ns) | Type | Slowest speed (nm ns$^{-1}$) | Smallest force (pN) | Size (#atoms) | Size (nm$^3$) |
|---|---|---|---|---|---|---|---|---|
| PCDH15 EC8–10 S | S1a–c | 4XHZ | 901.2 | EQ | — | — | 115,625 | 16 × 10 × 7.5 |
| PCDH15 EC8–10 B | S1d–e | 4XHZ | 56.1 | EQ | — | — | 204,024 | 28 × 10 × 7.5 |
| PCDH15 EC8–10 B | S2a–d | 4XHZ | 65.0 | PCV(COM) | 0.2 | — | 204,024 | 28 × 10 × 7.5 |
| PCDH15 EC8–10 B | S3a–d | 4XHZ | 1,512.7 | PCF | — | 10 | 204,024 | 28 × 10 × 7.5 |
| PCDH15 EC8–10 B | S4a–d | 4XHZ | 106.4 | EQ | — | — | 204,024 | 28 × 10 × 7.5 |
| PCDH15 EC8–10 B | S5a–e | 4XHZ | 885.2 | PCV | 0.02 | — | 204,024 | 28 × 10 × 7.5 |
| PCDH15 EC8–10 B | S6a–c | 4XHZ | 32.8 | PCV(COM) | 1.0 | — | 204,024 | 28 × 10 × 7.5 |
| PCDH15 EC8–10 B | S7a–d | 4XHZ | 361.9 | EQ | — | — | 204,024 | 28 × 10 × 7.5 |
| PCDH15 EC8–10 B | S8a–c | 4XHZ | 27.6 | PCV(NVE) | 1.0 | — | 204,024 | 28 × 10 × 7.5 |
| PCDH15 EC8–10 B | S9a–g | 4XHZ | 670.5 | EQ | — | — | 204,024 | 28 × 10 × 7.5 |
| DNcad EC2–3 | S10a–b | 3UBG | 51.21 | EQ | — | — | 134,046 | 22 × 7.1 × 8.9 |
| DNcad EC2–3 | S11a–c | 3UBG | 283.7 | PCF | — | 10 | 134,046 | 22 × 7.1 × 8.9 |
| DNcad EC2–3 | S12a–b | 3UBG | 151.2 | PCV | 0.1 | — | 134,046 | 22 × 7.1 × 8.9 |
| Chimeric complex | S13a–c | — | 61.2 | EQ | — | — | *209,952 | 44 × 6.9 × 7.3 |
| Chimeric complex | S14a–b | — | 177.5 | PCV | 0.1 | — | *209,952 | 44 × 6.9 × 7.3 |
| PCDH15 EC9–10$_{S5b-75ns}$ | S15a–b | — | 2,122.0 | EQ | — | — | 105,545 | 13 × 9.1 × 9.2 |
| PCDH15 EC9–10$_{S5b-76ns}$ | S16a–b | — | 7,351.0 | EQ | — | — | 98,124 | 13 × 9.2 × 8.8 |

Overview of cadherin MD simulations. Labels indicate the system and protein used (S: small system; B: big system for SMD). Type denotes the simulation protocol used (equilibrium: EQ, constant-velocity SMD: PCV, and constant-force SMD: PCF). COM denotes simulations in which forces were applied to the center of mass of multiple atoms (Supplementary Table 2). NVE indicates simulations performed in the microcanonical ensemble. Initial size of the systems is indicated in the last column. Simulations with the chimeric complex (indicated with an *) involve two systems encompassing 209,952 and 209,942 atoms.

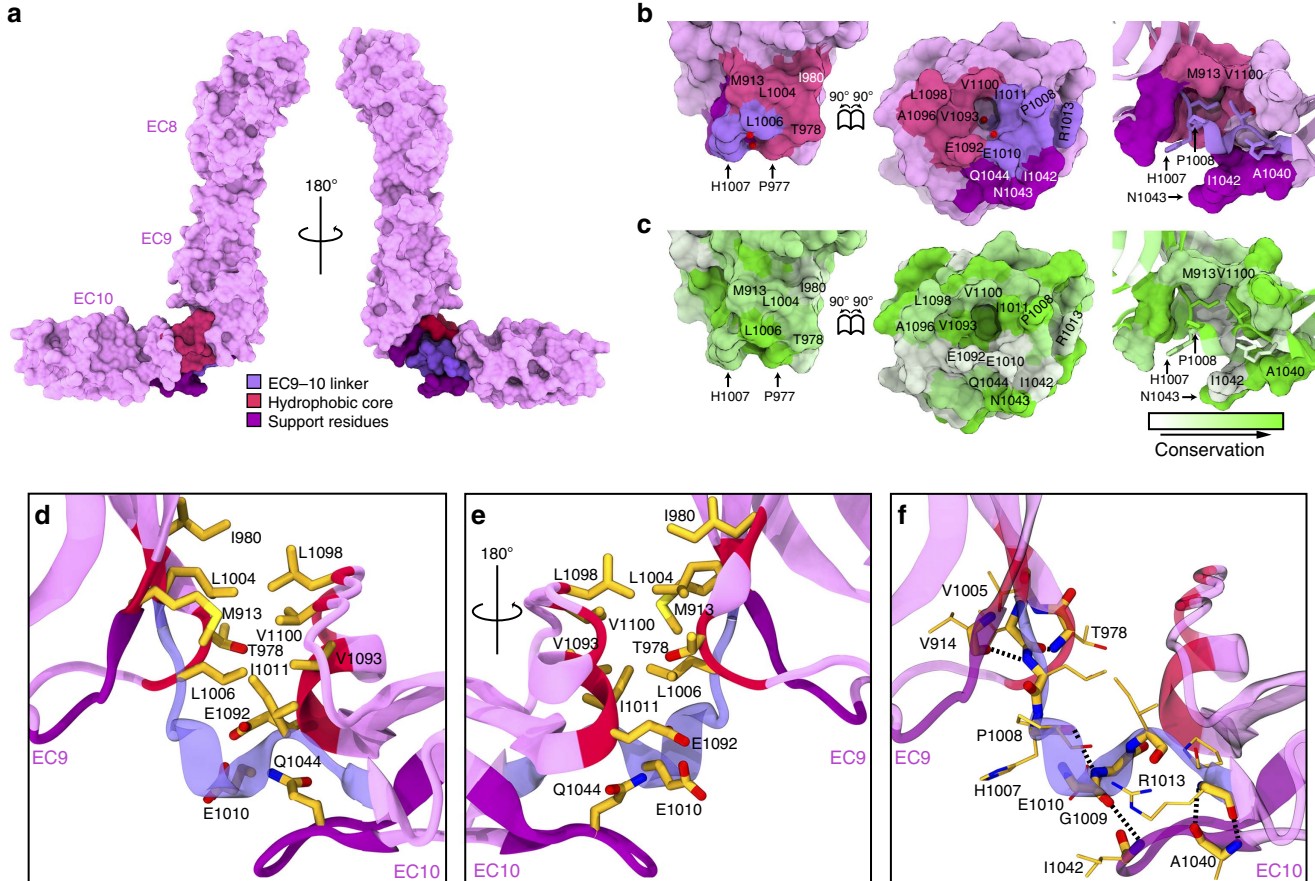

**Figure 3 | Structure of PCDH15 EC9–10 interface.** (**a**) Surface representation of PCDH15 EC8–10 (magenta). The EC9–10 interface is shown with its EC9–10 linker (violet), a hydrophobic core (dark pink) and supporting residues (purple). (**b**) PCDH15 EC9–10 interaction surfaces exposed (left and middle panels) and coloured as in **a** with interfacing residues labeled. Right panel shows side view as in **a**. Red spheres correspond to a pair of crystallographic water molecules. (**c**) Conservation of residues in the EC9–10 interface according to ConSurf[80] and the alignment in Supplementary Fig. 4. (**d,e**) Detail of the EC9–10 interface. Protein backbone is shown as ribbons and relevant residues are shown as sticks. Some backbone atoms are omitted for clarity. (**f**) Detail of backbone-hydrogen bonds (dashed lines) found in the EC9–10 linker.

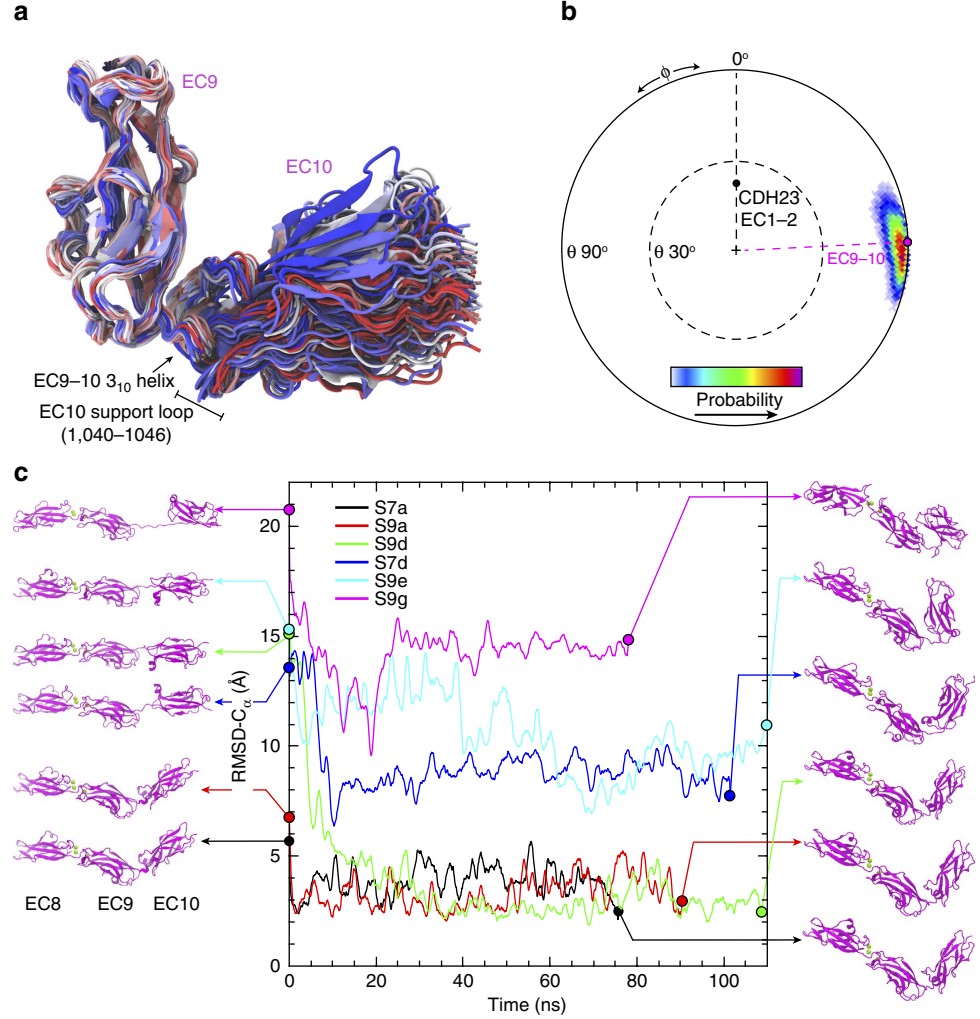

**Figure 4 | Simulated dynamics of PCDH15 EC9–10. (a)** Superposition of PCDH15 EC9–10 conformations taken every 5 ns from an equilibrium MD simulation and using EC9 as reference (S1c; Table 2). Colour indicates time (blue–white–red). The EC9–10 $3_{10}$ helix and EC10 support loop switch conformations throughout the trajectory. **(b)** Conformational freedom of EC10 with respect to EC9 as in Fig. 2e. The initial orientations for EC9–10 and CDH23 EC1–2 (2WHV, used as reference) are indicated. **(c)** r.m.s.d.-$C_\alpha$ from selected 'relaxation' equilibrium MD simulations (Movie 1) started from stretched conformations (left). Final conformations of each relaxation run are shown on the right. r.m.s.d.-$C_\alpha$ is shown as 1 ns running average for all cases.

PCDH15 EC10 with respect to EC9, monitored throughout the simulations, indicates some flexibility (Fig. 4a,b; Supplementary Fig. 5b), which is reflected in large values for the overall root-mean-square deviation of $C_\alpha$ atoms (r.m.s.d.-$C_\alpha$ between 4 and 6 Å), but stable values for individual EC repeats (<2.5 Å; Supplementary Figs 8a and 9a). In simulation S1b, a discrete jump in r.m.s.d.-$C_\alpha$ was observed for EC10 at ~300 ns, which correlates with a 'flip' in the conformation of the supporting loop (residues 1,040–1,046) following the rupture of a Gly 1,009:O–Ile 1,042:N hydrogen bond and with the partial unfolding of the EC9–10 $3_{10}$ helix (Fig. 3f; Supplementary Fig. 9a,c,d). This transition was not observed in simulation S1c (Supplementary Fig. 8a,c,d). Also, unbending was not observed and the end-to-end angle and distance were stable throughout both simulations (Supplementary Figs 8e,f and 9e,f). This suggests that the EC9–10 repeats are bent in solution, spontaneously and occasionally transitioning between two very similar bent states.

To further probe the bent conformation observed in the PCDH15 EC9–10 structure, we performed equilibrium MD simulations of partially stretched conformations of PCDH15 EC8–10 obtained from SMD simulations. Three sets of relaxations were performed. In the first set (S4 series), stretched states

were obtained from a fast constant velocity SMD simulation performed at 2 nm ns$^{-1}$ (S2a; $t = 3.8$, 4, 4.3 and 5 ns) and used as starting conformations for the new simulations (S4a–d). Similarly, in the second set (S9a–g series), stretched states were obtained from a slow constant-velocity SMD simulation performed at 0.1 nm ns$^{-1}$ (S5b, $t = 60$, 67, 70, 75, 76, 81 and 101 ns). In the last set (S7a–d), stretched states were obtained from a constant-force SMD simulation (100 pN, S3c; $t = 3$, 7.5, 10 and 26 ns). All relaxations lasted <100 ns. The native, bent conformation was recovered (r.m.s.d.-$C_\alpha$ <5 Å) for six out of 15 simulations (Fig. 4c; Supplementary Movie 1). In these cases, native hydrogen bonds and hydrophobic interactions were restored. In some simulations, relaxation of overly stretched states resulted in bent, twisted conformations in which EC10 was unable to fit back into EC9 as observed in our crystal structures. This suggests that re-bending (to a native conformation) of overly stretched and twisted states may occur in longer timescales, while unbending is easily reversible after small-scale stretching.

To explore stability and re-bending at longer timescales, two additional relaxations of stretched states of PCDH15 EC9–10 were performed using the Anton supercomputer[42]. The stretched states, obtained from simulation S5b (at 75 and 76 ns), were

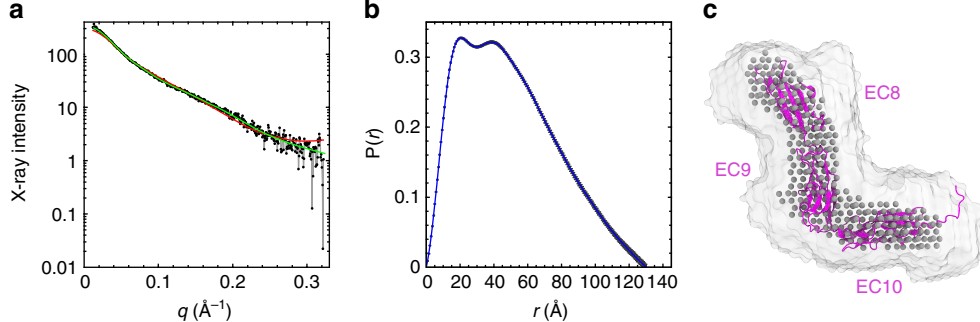

**Figure 5 | Low-resolution PCDH15 EC8–10 structure by SAXS. (a)** X-ray scattering intensity as a function of the scattering vector q (SAXS profile; black). Predicted scattering intensities from SAXS model ($\chi^2 = 1.01$; DAMMIF) and the PCDH15 EC8–10 structure ($\chi = 1.59$; FOXS) are shown as green and red lines, respectively. **(b)** Real-space pair distribution function ($P(r)$) from SAXS data. **(c)** Superposition of PCDH15 EC8–10 structure (magenta) with an averaged envelope (white surface) and a filtered solution (gray spheres) from *ab initio* models.

simulated for 2.2 and 7.3 µs, respectively (Supplementary Fig. 10a,b). As previously observed in simulation S9d (Fig. 4c), the S5b-75 ns stretched structure quickly re-bends to the native, EC9–10 crystallographic conformation with r.m.s.d.-$C_\alpha$ < 3 Å (S15b; Supplementary Fig. 10a). Interestingly, minor fluctuations in r.m.s.d.-$C_\alpha$ with minimal variations of end-to-end angle suggest again that PCDH15 EC9–10 can transition between two very similar bent conformations in solution. This simulation also confirms that a bent EC9–10 is stable over microseconds. In contrast, the S5b-76 ns stretched structure did not recover the native, crystallographic conformation, as indicated by large r.m.s.d.-$C_\alpha$ values (>8 Å) during 7.3 µs of dynamics (Supplementary Fig. 10b). However, it did adopt a conformation in which the repeat orientation and angle resembles the native, crystallographic state. Recovery of the native EC9–10 bent conformation from overly stretched and twisted states may require timescales >7 µs. Restricted conformational freedom in the context of an entire tip link might prevent twisting and favour faster re-bending.

**EC9–10 bent conformation is consistent with SAXS data.** To experimentally validate the structure of PCDH15 EC8–10 in solution, its molecular shape and particle dimensions were studied using small-angle X-ray scattering (SAXS). A representative SAXS profile is shown in Fig. 5a, from which estimates of the radius of gyration ($R_g$) can be obtained (Supplementary Table 3). Two alternate analyses indicate $R_g$ values of 37.1 ± 1.1 Å (Guinier analysis; Supplementary Fig. 11a) and 40.6 ± 0.2 Å (indirect Fourier transform of SAXS profile with maximum dimension $D_{max} = 130$ Å; Fig. 5b). Both estimates are in good agreement with each other and with the theoretical $R_g$ obtained from the PCDH15 EC8–10 structure (37 Å). This suggests that shape is maintained in solution.

An analysis using the Kratky and Kratky–Debye plots generated from SAXS data (Supplementary Fig. 11b,c) indicates that the protein is folded in solution, but it exhibits significant flexibility. This flexibility may arise from inter-repeat motion (Figs 2d,e and 4a,b; Supplementary Figs 8 and 9) or the presence of a C-terminal tail including EC11 residues and a His-tag used for protein purification. Despite this flexibility, *ab initio* modeling from the experimental scattering profiles produced PCDH15 EC8–10 shapes (in solution) that are in remarkable agreement with crystallographic data (Fig. 5c). Both the EC8–9 contortion and the EC9–10 kink can be observed in the filtered shape reconstruction showing an S-shaped molecule. Fits of scattering profiles obtained from the SAXS model and the PCDH15 EC8–10 structure are in good agreement with the SAXS data (green and red lines in Fig. 5a). Overall, these results indicate that the

PCDH15 EC8–10 crystallographic conformation, including bending at EC9–10, is also observed in solution.

**Forced unbending of PCDH15 EC8–10 *in silico*.** The function of tip links is to convey force and pull open inner ear transduction channels. To facilitate this process, myosin motors pull on tip links and keep a resting tension that maintains transduction channels optimally poised to respond to external mechanical stimuli[16,19]. In frogs, resting tension is likely to be ∼10 pN (∼5 pN per strand for a dimeric tip link), while external physiological stimuli could reach >100 pN (refs 13,14). The elasticity of the entire tip link is unknown, but previous studies on the elasticity of CDH23 suggest that its EC repeats are stiff and inextensible unless large forces are applied to unfold them[32]. The structure of PCDH15 EC8–10 suggests an alternate scenario where EC9–10 unbending might be relevant during force transduction: as force is applied to tip links, unbending results in protein extension without unfolding (tertiary structure elasticity: TSE[43,44]). If unbending requires low force (<10 pN), resting tension in the native tip link may keep PCDH15 EC9–10 unbent and TSE will not play a key role in mechanotransduction. If, on the other hand, intermediate forces (>10 pN) are required to unbend EC9–10, its TSE will affect channel gating.

To test the elastic response of PCDH15 EC8–10 we performed an extensive set of SMD simulations using constant-force and constant-velocity stretching approaches. In the constant-force simulations, the solvated EC8–10 repeats were stretched by applying a constant force to N- and C-terminal $C_\alpha$ atoms in opposite directions. Four simulations with forces of 10, 25, 50 and 100 pN were carried out (S3a, S3d, S3b and S3c; Table 2). Complete unbending of EC9–10 was observed as an end-to-end length increase of ∼4 nm for all simulations in which forces were >10 pN (Fig. 6a).

For the 10 pN simulation, lasting 1.03 µs, complete unbending was not observed (Fig. 6a–d). A sudden ∼2 nm increase in the end-to-end distance was observed at $t \approx 330$ ns, which lasted for ∼120 ns (Fig. 6a). This length increase correlates with the rupture of the Gly 1,009:O–Ile 1,042:N backbone hydrogen bond at the EC9–10 linker (Fig. 6b,c,e–h), which allows EC10 to rotate about its long principal axis and change the position of the helix in the FG-α loop (residues 1,091–1,095). The EC9–10 linker (Val 1,005:$C_\alpha$–Arg 1,013:$C_\alpha$) and a backbone hydrogen bond between His 1,007 and Glu 1,010 also extend, although more subtly. The EC10 rotation correlates with a decrease in the Leu 1,004–Ala 1,096 distance (Fig. 6c). As the simulation progressed with the 10 pN force applied, the native conformation was recovered at $t \approx 480$ ns along with the Gly 1,009:O–Ile 1,042:N hydrogen bond. Yet again, at $t \approx 880$ ns, a second stretching event

of ∼2 to ∼3 nm is observed, with the same displacement of the helix in the FG-α loop. Partial recovery of the native conformation is observed at $t > 1\,\mu s$ (Fig. 6h) and no unbending or EC10 unfolding were observed, as reported by interatomic distances Arg 1,013:O–Ala 1,040:N and Arg 1,013:N–Ala 1,040:O (Fig. 6b). A similar, albeit less pronounced, lengthening event was observed in one of the equilibrium MD simulation of the structure (Supplementary Fig. 9), suggesting that the EC9–10 linker can readily extend ∼1 nm and transition between the native, crystal conformation, and a slightly unbent conformation. Complete unbending is thus predicted to occur in two steps and only at forces $> 10\,pN$.

Similar constant-force simulations of the bent DNcad EC2–3 structure showed a different mechanical response. The EC2–3 linker is bent with an angle of ∼80° (Supplementary Fig. 6c)[33] and residues expected to form calcium-binding motifs are replaced by non-charged ones that form an interface with a buried surface area of 398 Å² (Supplementary Fig. 6d). For PCDH15 EC9–10 a total of 16 residues participate on the interface with a buried surface area of 389 Å². Yet, the DNcad EC2–3 interface does not have features similar to those found in PCDH15 EC9–10 (FG-α loop and EC9–10 $3_{10}$ helix), and it has a larger fraction of polar residues compared with the more hydrophobic PCDH15 EC9–10 interface. Consistently, the

DNcad EC2–3 structure unbends fully and quickly (∼150 ns) in simulations with an applied constant force of 10 pN (Supplementary Fig. 6e,f), and application of larger forces shows even faster unbending (similar results were obtained when doing constant-velocity simulations, Supplementary Fig. 6g,h). Taken together, the PCDH15 EC8–10 and DNcad EC2–3 simulations suggest that the bent EC9–10 interface is unique and able to withstand small forces without unbending in a microsecond timescale. Therefore, its TSE is predicted to be relevant during mechanotransduction.

**Forced unfolding of PCDH15 EC8–10** *in silico*. To further explore the elasticity of PCDH15 EC8–10, we carried out constant-velocity SMD simulations in which both protein ends are attached to springs that are pulled in opposite directions. The applied force throughout the simulation is obtained from each spring's extension. The PCDH15 EC8–10 equilibrated native state was stretched at speeds that ranged from 0.02 to 10 nm ns⁻¹ (simulations S2a–d, S5a–e, S6a–c, S8a–c; Table 2; Supplementary Movie 2). At fast pulling speeds (5 and 10 nm ns⁻¹), force increased rapidly (Fig. 7a), with unbending quickly followed by unfolding of EC10 at the peak force. For the slower stretching speeds, force barely increased during unbending and straightening of the EC9–10 linker (phase I), with the structure

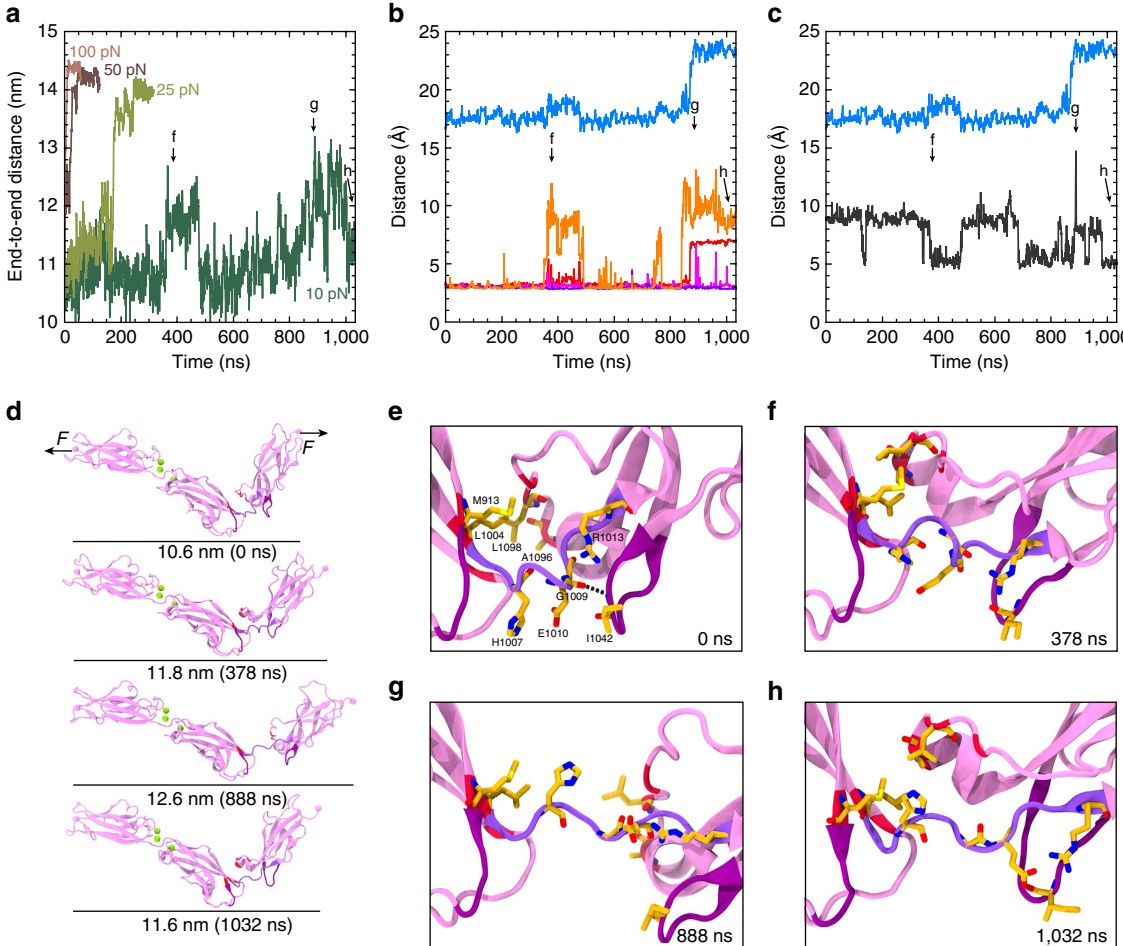

**Figure 6 | Constant-force SMD simulations of PCDH15 EC8–10.** (**a**) End-to-end distance measured for constant-force stretching of PCDH15 EC8–10 at 10 pN (simulation S3a, olive green), 25 pN (S3d, green), 50 pN (S3b, dark brown) and 100 pN (S3c, light brown). A 1 ns running average is shown in all cases. Interatomic distances for: (**b**) Val 1,005:$C_\alpha$–Arg 1,013:$C_\alpha$ (light blue) His 1,007:O–Glu 1,010:N (red), Gly 1,009:O–Ile 1,042:N (orange), Arg 1,013:O–Ala 1,040:N (magenta) and Arg 1,013:N–Ala 1,040:O (purple), and (**c**) Leu 1,004:$C_\gamma$–Ala 1,096:$C_\beta$ (dark gray). (**d**) Snapshots of initial conformation and mechanically induced unbending of PCDH15 EC8–10 taken from the 10 pN simulation (S3a). Protein is shown as ribbons and coloured as in Fig. 3. Arrows indicate position and direction of the applied forces. (**e–h**) Detail of the EC9–10 linker at time points indicated in **a–c**.

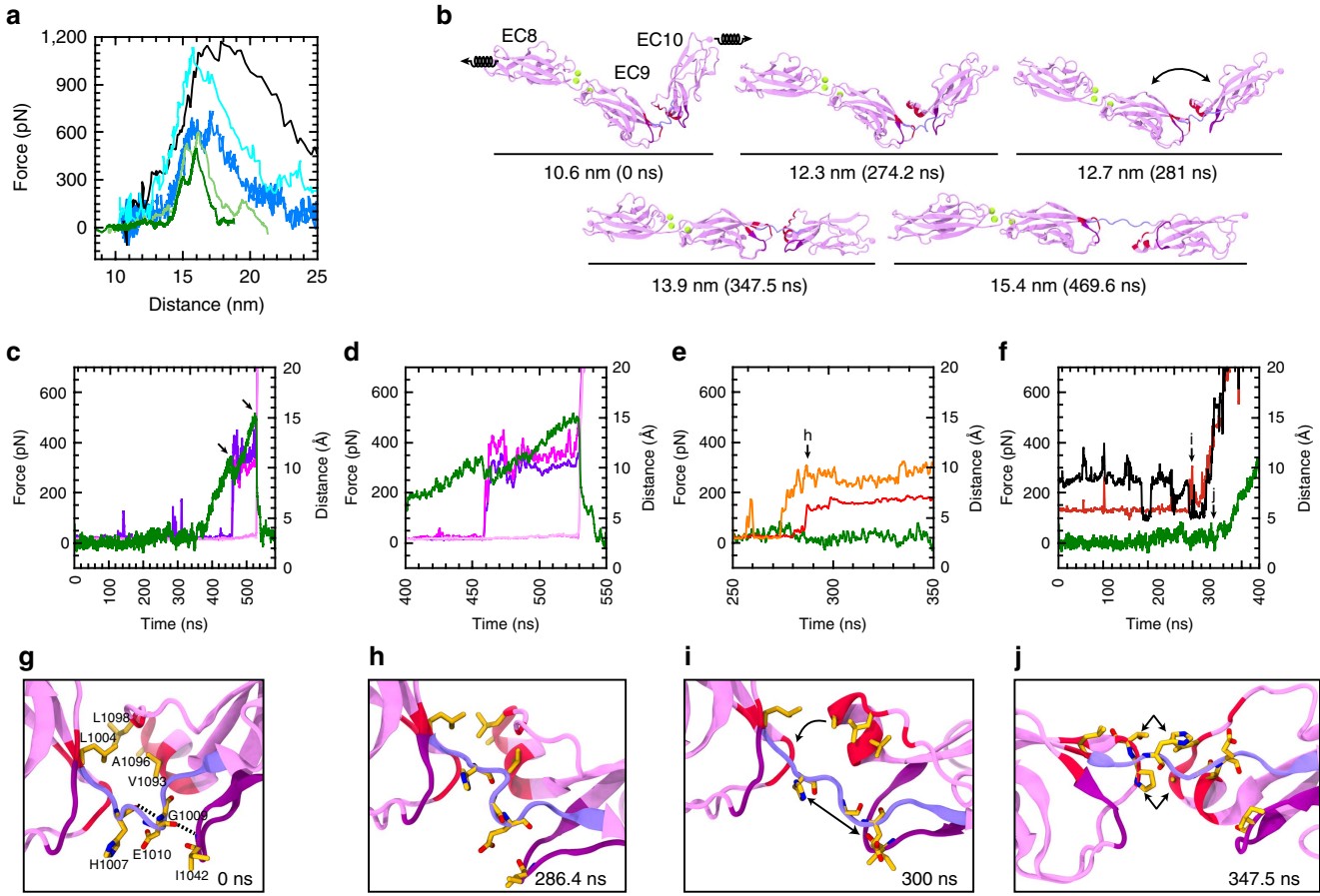

**Figure 7 | Constant-velocity SMD simulations of PCDH15 EC8–10. (a)** Force applied to N terminus versus end-to-end distance for constant-velocity stretching of PCDH15 EC8–10 at 10 nm ns$^{-1}$ (simulation S5d, black), 5 nm ns$^{-1}$ (S5c, turquoise), 1 nm ns$^{-1}$ (S5a1, light blue), 0.1 nm ns$^{-1}$ (S5b, 1 ns running average, light green) and 0.02 nm ns$^{-1}$ (S5e, 1 ns running average, dark green). **(b)** Snapshots of initial conformation and mechanically induced unbending and unfolding of PCDH15 EC8–10 taken from the 0.02 nm ns$^{-1}$ simulation (S5e; Movie 2). Protein is shown in ribbon representation and coloured as in Fig. 3. Springs indicate position and direction of the applied forces. **(c–f)** Force applied to PCDH15 EC8–10 N terminus (S5e, dark green) along with interatomic distances for **(c,d)** Arg 1,013:O–Ala 1,040:N (magenta) and Arg 1,013:N–Ala 1,040:O (purple); Tyr 1,019:N–Lys 1,108:O (light magenta); and Tyr 1,019:O–Tyr 1,110:N (pink). Rupture of these interactions correlates with unfolding force peaks. **(e)** Interatomic distances for His 1,007:O–Glu 1,010:N (red), Gly 1,009:O–Ile 1,042:N (orange), **(f)** Leu 1,004:C$_\gamma$–Leu 1,098:C$_\gamma$ (maroon), and Leu 1,004:C$_\gamma$–Ala 1,096C$_\beta$ (dark gray). Rupture of these interactions correlates with unbending. A 1 ns running average is shown in all cases. **(g–j)** Snapshots of the EC9–10 linker during S5e at time points indicated in **e,f**.

lengthening by ∼4 nm before forces started to increase rapidly (phase II) until unfolding occurred (Fig. 7a–f; Supplementary Fig. 12a). The observation of these two phases suggests that unbending is important for the elastic response of PCDH15.

Unbending of PCDH15 EC9–10 during the slowest constant-velocity stretching simulation was similar to unbending in constant-force mode. Hydrophobic interactions were re-accommodated as the EC9–10 linker was stretched, with rupture of the hydrogen bonds that form the EC9–10 3$_{10}$ helix (His 1,007:O–Glu 1,010:N and Pro 1,008:O–Ile 1,011:N) and the backbone hydrogen bond Gly 1,009:O–Ile 1,042:N (Fig. 7e). The helix in the FG-α loop of EC10 (formed by residues 1,091–1,095) rotates towards EC9 at $t \approx 280$ ns (12.7 nm; Fig. 7g–i) with a decrease in the distance between Leu 1,004:C$_\gamma$ and Ala 1,096:C$_\beta$ lasting ∼50 ns (Fig. 7b,f,j). At this point, the EC9–10 linker was fully extended, but a few contacts between the two EC repeats remain. As the simulation continued, applied force increased linearly reaching >300 pN with little protein extension until the rupture of a pair of backbone hydrogen bonds (Arg 1,013:O–Ala 1,040:N and Arg 1,013:N–Ala 1,040:O; Fig. 7a,c,d). These events were accompanied by a sudden drop in applied force and marked the beginning of EC10 unfolding. A second

force peak correlated with the rupture of backbone hydrogen bonds Tyr 1,019–Lys 1,108 and Tyr 1,019–Tyr 1,110 (Fig. 7a,c,d). The maximum force peak associated to unfolding for all constant-velocity simulations was always >400 pN (Supplementary Fig. 12a).

Interestingly, a salt bridge between residues Glu 1,010 and Arg 1,013 formed during the slow speed and low force stretching simulations of PCDH15 EC8–10 (Supplementary Fig. 13). These residues are part of the EC9–10 linker and do not interact with each other in the native, crystallographic conformation. However, this salt bridge is intermittently formed in one of our equilibrium simulation (S1b) as a consequence of the relaxation of the EC9–10 3$_{10}$ helix and a conformational change of the EC10 supporting loop (1,040–1,046) discussed above (Supplementary Fig. 9g). Formation of the Glu 1,010–Arg 1,013 salt bridge may help prevent unfolding by strengthening the EC9–10 linker in its extended conformation.

To quantify the stiffness of PCDH15 EC8–10 we computed the slope of the linear regions of the force versus end-to-end distance monitored in simulations using the slowest pulling velocities (S5b at 0.1 nm ns$^{-1}$ and S5e at 0.02 nm ns$^{-1}$). During unbending (phase I), the effective spring constants were $k_{S5b-I} = 30$ mN m$^{-1}$ and $k_{S5e-I} = 8.4$ mN m$^{-1}$. After unbending but before

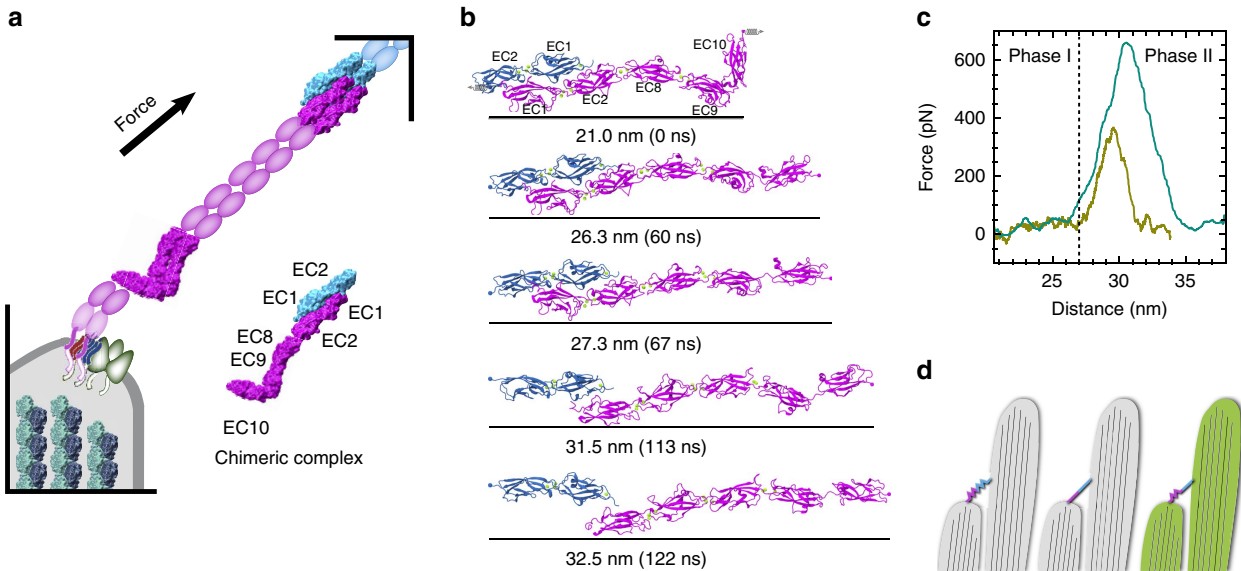

**Figure 8 | Mechanics of a chimeric tip link.** (**a**) Transduction apparatus diagram. The full-length PCDH15 (magenta), with its 11 EC repeats, forms a homophilic dimer that interacts with CDH23 (light blue). Known structures, including EC8–10, are shown. Inset shows a chimeric complex with CDH23 EC1–2 interacting with PCDH15 EC1-2-8-10. (**b**) Snapshots of initial conformation and mechanically induced unbending and unbinding of the chimeric complex taken from the 0.1 nm ns$^{-1}$ simulation (S14a; Movie 3). Springs indicate position and direction of the applied forces. Unbinding with minimal unfolding follows unbending. (**c**) Force applied to one of the C termini versus end-to-end distance of the complex for constant velocity stretching at 1 (S14b1; teal) and 0.1 nm ns$^{-1}$ (S14a; olive). (**d**) Models of tip-link function as a gating spring (left), stiff cable (middle) and a semi-elastic filament (right).

unfolding (phase II), we obtained $k_{S5b-II} = 445$ mN m$^{-1}$ and $k_{S5e-II} = 320$ mN m$^{-1}$. In contrast, two straight CDH23 EC repeats lack the first, soft, phase during stretching and their elasticity is 570 mN m$^{-1}$ at 0.1 nm ns$^{-1}$ (ref. 32). These results suggest that the PCDH15 EC8–10 fragment behaves as a soft spring during unbinding (phase I) of the EC9–10 linker.

Additional SMD simulations performed in the *NVE* ensemble (S8 series in Table 2 and Supplementary Table 2) or by pulling on the center-of-mass of multiple C$_\alpha$ atoms at each end of PCDH15 EC8–10 (S2 and S6 series) revealed similar scenarios for unbending and unfolding.

**Unbending and unbinding in an artificial chimeric tip link.** To understand the elastic response of the PCDH15 EC8–10 fragment in the context of the entire tip link, we designed a 'chimeric' model involving all known structures of PCDH15 and CDH23 EC repeats (Fig. 8a). The structure of the tip-link handshake including CDH23 EC1–2 and PCDH15 EC1–2 was coupled to PCDH15 EC8–10 via an artificial PCDH15 EC2–8 canonical linker (Supplementary Fig. 14a). The resulting PCDH15 EC1–2–8–10 + CDH23 EC1–2 chimera, including seven EC repeats, allowed us to test *in silico* whether unbending and unfolding of PCDH15 EC9–10 could occur before unbinding of PCDH15 from CDH23.

Constant-velocity SMD simulations of the chimeric complex using stretching speeds of 0.1 and 1 nm ns$^{-1}$ (simulations S14a and S14b1–3, respectively; Table 2; Supplementary Table 2) showed unbending of EC9–10 followed by unbinding of PCDH15 from CDH23 without significant unfolding of any of the EC repeats (Fig. 8b; Supplementary Movie 3; Supplementary Fig. 14c–f). Force monitored throughout these simulations (Fig. 8c; Supplementary Figs 12b and 14b) revealed the same two phases observed for PCDH15 EC8–10 alone. Unbending during a 7 nm extension of the complex (phase I) required low forces (<100 pN) with an effective stiffness of $k_{S14a-I} = 8.4$ mN m$^{-1}$ at 0.1 nm ns$^{-1}$ (compared with $k_{S5b-I} = 30$ mN m$^{-1}$ for EC8–10 at the same speed). Stretching

at slower speeds of longer tip links may reveal an even smaller effective spring constant. Stretching of the straightened structure (phase II) resulted in a rapid increase of the applied force ($k_{S14a-II} = 142$ mN m$^{-1}$) until unbinding happened at forces >400 pN (Fig. 8c; Supplementary Figs 12b and 14b). Unfolding of EC10 required more force (794.4 ± 107.9 pN and 650 pN at 1 and 0.1 nm ns$^{-1}$, respectively) than the unbinding of the chimeric complex (651.6 ± 41.6 pN and 420 pN at 1 and 0.1 nm ns$^{-1}$; Supplementary Fig. 12). These simulations suggest that the PCDH15 EC9–10 linker does not unfold and may unbend before unbinding during tip-link-mediated mechanotransduction.

## Discussion

The structural studies and simulations of PCDH15 presented here provide a first view into its non-canonical cadherin EC repeats and their function as part of inner-ear tip links. Our data suggest that EC9–10 is bent in solution, that TSE stemming from unbinding is relevant for hair-cell mechanotransduction, and that PCDH15-CDH23 unbinding occurs before unfolding of individual EC repeats of known structure.

The most striking feature of the PCDH15 EC8–10 structure is the bent conformation adopted by the calcium-free EC9–10 linker. Multiple lines of evidence suggest that this conformation is particular and favoured in solution. First, the lack of calcium-binding motifs sequences at the EC9–10 linker is highly conserved across different species. In addition, EC9 and EC10 have unusual and unique structural features involved in bending, namely an EC9–10 3$_{10}$ helix and an EC10 FG-α loop that form part of an interface that keeps EC9 stuck to EC10. These features are also conserved with only subtle differences between mammalian, aves and reptilian, amphibian and actinopterygian sequences. Last, short and long timescale MD simulations as well as SAXS experiments strongly support a bent and somewhat flexible conformation in solution. Interestingly, the highest resolution views of tip links *in situ* show what could be interpreted as a bent EC9–10 linker near the point of membrane insertion of PCDH15 (Fig. 3d in Kachar *et al.*[6]).

The existence of the EC9–10 bent conformation has important implications for PCDH15 function in sensory transduction. Tip links are constantly subjected to forces that may range from 1 to >100 pN (refs 12–14), and force-induced transitions from bent to straightened conformations might regulate transduction-channel gating and adaptation. SMD simulations predict that unbending of EC9–10 does not happen at 10 pN, suggesting that this bent conformation will exist in native tip links, even when myosin motors are applying a resting tension of ∼10 pN to them (∼5 pN per strand for a dimeric tip link). We note that EC11 and a straight EC10–EC11 linker would provide a longer 'lever arm' that would result in a larger applied torque at the same force, which may favour unbending at smaller forces. However, the human EC10–11 linker lacks one of the residues involved in calcium binding (Supplementary Fig. 2), and might have enhanced flexibility that would hamper the action of a longer lever arm (Supplementary Fig. 15). Unbending transitions provide a soft elasticity that resembles TSE predicted for ankyrin repeats[43] and titin domains[44]. Small, but not negligible force is required to reversibly straighten these proteins without unfolding, a process that will affect how force is conveyed to transduction channels.

Biophysical experiments predict the existence of a soft hair-cell 'gating spring' with an elasticity of ∼1 mN m$^{-1}$, a value estimated from measurements in timescales of several hundreds of microseconds[12,14]. While ultra-structural studies of tip links[6] and simulations of CDH23 EC1–2 (ref. 32) suggest that tip links are stiff, our data indicates that TSE in PCDH15 will render the lower end of tip links somewhat elastic (Fig. 8d). Constant-velocity simulations predict an effective spring constant of ∼8 mN m$^{-1}$ with a working distance of ∼5 nm originating from EC9–10 unbending TSE. In the context of a dimeric tip link, where parallel unbending of two EC9–10 repeats would render PCDH15 stiffer (>16 mN m$^{-1}$), either other parts of PCDH15 or other components of the transduction apparatus must provide the missing elasticity. However, the spring constant estimates from simulations represent an upper boundary to physiological values, as stretching speeds *in silico* are significantly faster than those used in the experiments that were used to estimate the gating spring constant (Supplementary Discussion).

In PCDH15, the EC5–6 linker lacks calcium-binding residues and might unfold, be flexible, or adopt a bent conformation as well[33,38,45,46]. Also, a possible EC12 repeat and its linker to EC11 could provide additional elasticity along with the lipid membrane around the transduction channel[47,48]. Electrophysiological measurements suggest that if the tip link is the gating spring, it might stretch by 200 nm for extreme mechanical deflection of hair bundles[49]. Unbending only provides a 5 nm extension, implying that such large stretching would either require unfolding of EC repeats or membrane tether formation at the tip link insertion point[50,51]. Regardless of the mechanism, the elastic response of the whole transduction apparatus will be dominated by the softest component, yet to be determined. Clearly, more complete structural models of the hair-cell transduction apparatus are required to dissect out the exact contribution of each of its components to the gating spring elasticity, with unbending TSE of PCDH15 EC9–10 being part of it.

Unbending of PCDH15 EC9–10 could also regulate adaptation in hair-cell mechanotransduction. In constant-force simulations, unbending results in a ∼5 nm lengthening of the structure at forces >10 pN. Tip link lengthening due to unbending could lead to decreased tip link tension and channel closure. In this speculative scenario, PCDH15 EC9–10 would play a role similar to that of a hypothetical 'release element' thought to mediate fast, calcium-dependent adaptation[52,53]. Calcium would have to regulate this process allosterically through other PCDH15 and

CDH23 EC repeats with calcium-binding motifs, thereby introducing a non-linear response and modulating the stiffness of the release element as proposed by Martin et al.[53].

In addition to unbending, unfolding of PCDH15 EC repeats can also drastically alter how force is conveyed to transduction channels. Calcium is known to help prevent mechanical unfolding in cadherins[36,54], and mutations that alter cadherin calcium-coordinating residues cause deafness (see Supplementary Discussion and Supplementary Fig. 16). A calcium-free inter-repeat linker in PCDH15 could be prone to unfolding[32]. Constant-velocity SMD simulations of PCDH15 EC8-10 predict unfolding forces of ∼440 pN at 0.02 nm ns$^{-1}$ (Supplementary Fig. 14f), smaller than those monitored for unfolding of CDH23 EC1–2 repeats with bound calcium ions, but similar to those predicted for CDH23-PCDH15 unbinding[27,32]. Simulations of an artificial chimeric model involving all published structures of tip link EC repeats suggest that unbending of EC9–10 is followed by unbinding without any relevant unfolding, as observed for other classical cadherins[55,56]. However, this artificial model does not take into account the effect of parallel dimerization (see below) and does not incorporate the behaviour of other EC repeats and atypical linkers that may unfold before unbinding occurs.

Relating the nanomechanics of tip link structures to microscopic correlates of hair-cell bundle motion and channel gating obtained from biophysical experiments is challenging[47,48,57]. Tip links are enormous proteins, and structures for many EC repeats are still missing. While most CDH23 EC repeats are predicted to be canonical, several PCDH15 EC repeats have atypical sequence features that may alter their structure and function. In addition, tip links are supposed to be made by parallel dimers of PCDH15 and CDH23 interacting tip-to-tip ('parallel' is defined here to describe two proteins with their N- to C-termini directions aligned). Parallel dimerization may further alter, perhaps in trivial ways, the mechanical response of the whole assembly[58]. Glycosylation could also modulate dimerization and elasticity of PCDH15 (ref. 59) (see Supplementary Discussion and Supplementary Fig. 17). Moreover, while sound transduction is a fast process that can occur in microseconds, SMD simulations are performed at stretching velocities that match only part of the spectrum of physiological stimuli and that may not completely capture the viscoelastic response[60] of tip link cadherins (see Supplementary Discussion). Yet our structure and simulations of PCDH15 EC8–10 points to a key mechanical component of the tip link that is elastic and that must be incorporated in hair-cell transduction models.

Overall, the data presented here strongly suggest that the bent, calcium-free linker in PCDH15 EC9–10 plays a relevant mechanical role in hair-cell transduction. Our findings may have implications beyond PCDH15 function in the inner ear. Sequence analyses have shown that calcium-free linkers are found in multiple members of the cadherin superfamily, yet their function is unknown. Lack of a common sequence motif among these calcium-free linkers (except for the absence of calcium-binding residues) suggests that they could provide a variety of conformations and play diverse roles in cadherin function[33]. Perhaps some of these linkers endow non-classical cadherins with specific mechanical functions beyond the classical paradigm[59,61].

## Methods
**Expression and purification of PCDH15 EC8–10.** Human protocadherin-15 repeats EC8-10 (PCDH15 EC8–10) and mouse protocadherin-15 repeats EC9–10 (mmPcdh15 EC9–10), were subcloned into NdeI and XhoI sites of the pET21a vector. These PCDH15 repeats were expressed in BL21CodonPlus (DE3)-RIPL cells (Stratagene) cultured in TB and induced at $OD_{600} = 0.6$ with

1 mM of IPTG at 37 °C for ~16 h. Cells were lysed by sonication in denaturing buffer (20 mM TrisHCl (pH 7.5), 6 M guanidine hydrochloride, 10 mM CaCl$_2$ and 20 mM imidazole). The cleared lysates were loaded onto Ni-Sepharose (GE Healthcare), eluted with denaturing buffer supplemented with 500 mM imidazole and refolded by overnight dialysis against 20 mM TrisHCl (pH 8.0), 150 mM KCl, 400 mM arginine and 5 mM CaCl$_2$ using MWCO 2,000 membranes. Refolded protein was further purified on a Superdex200 column (GE Healthcare) in 20 mM TrisHCl (pH 8.0), 150 mM KCl and 5 mM CaCl$_2$ and concentrated by ultrafiltration to 2 mg ml$^{-1}$ for crystallization (Vivaspin 10 KDa).

**Crystallization and structure determination.** Crystals were grown by vapor diffusion at 4 °C by mixing 1 volume of protein and 0.5 volumes of reservoir solution (0.1 M MES (pH 6.5), 1.6 M NaCl for PCDH15 EC8–10, and 0.095 M HEPES-Na (pH 7.5), 0.19 M CaCl$_2$, 26.6% v/v PEG 400, 5% v/v glycerol anhydrous for mmPcdh15 EC9–10). Crystals of PCDH15 EC8–10 were cryoprotected in reservoir solution plus 25% glycerol. All crystals were cryo-cooled in liquid N$_2$. X-ray diffraction data was collected as indicated in Table 1 and processed with HKL2000. Both structures were determined by molecular replacement using PHASER[62]. A separate homology model for each repeat was used as initial search model for the PCDH15 EC8–10, while the mmPcdh15 EC9–10 structure was determined using EC9–10 from the PCDH15 EC8–10 structure. Model building was done with COOT[63] and restrained TLS refinement was performed with REFMAC5 (ref. 64). The mmPcdh15 EC9–10 structure was indexed in space group P 3$_1$ after detecting tetatohedral twinning. Refinement for this structure used the 'Twin Refinement' option in REFMAC5 after achieving $R_{work}$ and $R_{free}$ values ~40%. Data collection and refinement statistics are provided in Table 1.

**Small-angle X-ray scattering analysis.** SAXS data was collected at the SIBYLS beamline in the Advanced Light Source facility (Berkeley, CA) as described[65]. Three different concentrations of purified PCDH15 EC8–10 were analysed: 1, 1.5, 2 mg ml$^{-1}$. Higher concentrations were not tested due to protein aggregation observed above 2 mg ml$^{-1}$. Data were collected in 20 mM TrisHCl (pH 8.0), 150 mM KCl and 5 mM CaCl$_2$ at 20 °C and using four exposure times of 0.5, 1, 2 and 5 s. Buffer matched controls were used for buffer subtraction. Data analysis was carried out with PRIMUS[66] and the ATSAS program suite. Estimates of the radius of gyration ($R_g$) from the Guinier region were measured with PRIMUS. Maximum dimension ($D_{max}$) of particles was estimated from an indirect Fourier transform of the SAXS profiles using GNOM[67]. Values of $D_{max}$ between 120 and 140 Å provided the best solutions. *Ab initio* modeling was carried out with DAMMIF[68] using results from GNOM and considering $q < 0.2$. Fifteen models were generated and averaged, and a filtered envelope was produced with DAMAVER[69] with a mean normalized spatial discrepancy (NSD) of 1.04 ± 0.06 (one model rejected). Model scattering intensities were computed from PCDH15 EC8–10 (4XHZ) using FoXS[70].

**Simulated systems.** The psfgen, solvate and autoionize VMD[71] plugins were used to build all systems (Table 2; Supplementary Table 2). The structure of PCDH15 EC8–10 used for simulations consists of residues 795 to 1,116. The structure of DNcad EC2–3 (PDB code 3UBG) comprises residues 540 to 747. Hydrogen atoms were automatically added to protein structures and crystallographic water molecules. Residues D, E, K and R were assumed to be charged. Histidine residues were assumed neutral, and their protonation state was chosen to favour the formation of evident hydrogen bonds. Additional water molecules and randomly placed ions were used to solvate the systems at the desired ion concentration (150 mM KCl for PCDH15 EC8–10 and 150 mM NaCl for DNcad EC2–3).

To perform long timescale MD simulations using the Anton supercomputer (see below) a special set of systems was built to adhere to size limitations (<120,000 atoms). Conformations of stretched PCDH15 EC8–10 were taken from SMD simulation S5b at 75 and 76 ns. Two new structures were generated: PCDH15 EC9–10$_{S5b-75ns}$ and PCDH15 EC9–10$_{S5b-76ns}$ comprising residues 899 to 1,116 corresponding to PCDH15 EC9–10 only. These two structures where solvated and ionized as mentioned above.

The chimeric complex was constructed by linking together the tip-link handshake (CDH23 EC1–2 + PCDH15 EC1–2; 4APX) and PCDH15 EC8–10 (4XHZ, Fig. 8a). The artificial EC2–8 linker was created based on the high sequence similarity between EC repeats at the linker regions and using truncated and aligned structures of PCDH15 (EC1–2 residues 1–232, EC7–8 residues 787–796 and EC8–10 residues 797–1,116). The resulting model for the linker was energy-minimized in vacuum (2,000 steps), and subsequently analysed with the CheckMyMetal server[72] to ensure proper coordination of calcium ions (Supplementary Fig. 14a). Coordinates of the entire chimeric complex are available upon request.

**Molecular dynamics simulations using NAMD.** MD simulations were performed using NAMD 2.10 and 2.11 (ref. 73), the CHARMM36 force field for proteins with the CMAP correction and the TIP3P model for water[74]. A cutoff of 12 Å (with a switching function starting at 10 Å) was used for van der Waals interactions along with periodic boundary conditions. The Particle Mesh Ewald method was

used to compute long-range electrostatic forces without cutoff and with a grid point density of $>1$ Å$^{-3}$. A uniform 2 fs integration time step was used together with SHAKE. Langevin dynamics was utilized to enforce constant temperature ($T = 300$ K) when indicated, with a damping coefficient of 0.1 ps$^{-1}$ unless otherwise stated. Constant pressure simulations ($NpT$) at 1 atm were conducted using the hybrid Nosé-Hoover Langevin piston method with a 200 fs decay period and a 50 fs damping time constant. In one equilibrium simulation (S1c) the C$_\alpha$ atoms of residues 797, 833, 876 and 888 of EC8 were restrained ($k_r = 1$ kcal mol$^{-1}$ Å$^{-2}$) to avoid motions that may result in clashes between neighboring periodic images.

**Molecular dynamics simulations using Anton.** Anton is a massively parallel special purpose machine for molecular dynamics simulations[42]. Systems (PCDH15 EC9–10$_{S5b-75ns}$ and PCDH15 EC9–10$_{S5b-76ns}$) pre-equilibrated using NAMD (S15a and S16a; Supplementary Table 2) were converted to the Anton-compatible Maestro format using the convertNAMDtoDMS.py script provided by NRBSC/PSC. Anton and NAMD simulations used the same force field. Simulations were performed in the $NpT$ (300 K, 1 atm) ensemble with the Multigrator integration framework. Infrequent updates of both thermostat and barostat improve the performance of the simulation and numerical integration accuracy. In our simulation protocol, the Langevin thermostat[75] was updated every 24 steps and the MTK barostat was updated every 240 steps. The integration time step was set to 2 fs and frames were saved every 240 ps. Long-range electrostatic interactions were calculated with the k-Gaussian split Ewald method with a 64 × 64 × 64 grid. Accurate cut off values ranged between 10–13 Å and were automatically calculated by the Anton setup protocol based on the chemical features of the systems. SHAKE was used to constrain all bonds involving hydrogen atoms.

**Simulations and analysis tools.** Each system was energy-minimized and equilibrated in the constant number, pressure and temperature ensemble ($NpT$), and the resulting state was used to perform subsequent equilibrium and SMD simulations (Table 2). Constant-velocity stretching simulations used the SMD method and the NAMD Tcl forces interface. Constant-velocity SMD simulations[76–79] were performed by attaching C$_\alpha$ atoms of N- and C-terminal residues to independent virtual springs of stiffness $k_s = 1$ kcal mol$^{-1}$ Å$^{-2}$, or where indicated, by attaching the center of mass (COM) of groups of C$_\alpha$ atoms to the same type of virtual springs. The stretching direction was set along the $x$ axis matching the vector connecting terminal regions of the protein. The free ends of the springs were moved away from the protein in opposite directions at a constant velocity. Applied forces were computed using the extension of the virtual springs. Plotted forces correspond to those applied to the N-terminal atoms unless otherwise stated. Stiffness was computed through linear regression fits of force-distance plots. Maximum force peaks and their averages were computed from 50 ps running averages used to eliminate local fluctuations. Average and standard deviation of force versus distance curves were obtained by computing these quantities using data grouped in 1-Å bins. In constant-force SMD simulations, end-to-end or COM-to-COM distances were computed as the distances between individual SMD atoms or center-of-mass of SMD atoms at opposite protein ends, respectively. Principal axes of EC repeats were computed using the Orient VMD plugin. Sequence alignments were performed with MUSCLE. Plots and curve fits were prepared with QtiPlot. Molecular images were created with the molecular graphics program VMD[71].

**Data availability.** Coordinates for PCDH15 EC8–10 and Pcdh15 EC9–10 have been deposited in the Protein Data Bank with entry codes 4XHZ and 5KJ4. Diffraction images were deposited in the SBGrid Data Bank with entry codes 139 and 331. The remaining data that supports the findings of this study are available from the corresponding author upon reasonable request.

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

## Acknowledgements

We thank B. Derfler (D.P. Corey's laboratory) and R.E. Powers (R. Gaudet's laboratory) for assistance with initial cloning and crystallization experiments, the CCP4/APS school/workshop for training (to R.A.-S.), and G. Hura (SIBYLS beamline) for guidance on the analysis of SAXS data. This work was supported by the Ohio State University, by the National Institutes of Health (NIH; grants K99/R00 DC012534 to M.S.; P41GM103712-S1 to National Resource for Biomedical Supercomputing/Pittsburgh Supercomputing Center), and by the National Science Foundation through XSEDE (XRAC MCB140226). Simulations were performed at the TACC-Stampede, PSC-Anton, and OSC-Oakley (PAS1037) supercomputers. Use of APS NE-CAT beamlines was supported by NIH (P41 GM103403 & S10 RR029205) and the Department of Energy (DE-AC02-06CH11357) through grant GUP 40277. SAXS analyses were supported by the Department of Energy project Integrated Diffraction Analysis Technologies. R.A.-S. is a Pelotonia fellow and M.S. is an Alfred P. Sloan fellow (FR-2015-65794).

## Author contributions

M.S. and R.A.-S. performed the cloning, protein expression, purification, crystallization and solution of the X-ray structure. R.A.-S. prepared the samples and analysed the SAXS data. B.L.N. built the chimeric complex. R.A.-S., M.S. and B.L.N. designed, performed and analysed the molecular dynamics simulations. R.A-S., B.L.N. and M.S. wrote and edited the manuscript.

## Additional information

**Competing financial interests:** The authors declare no competing financial interests.

**Publisher's note**: 

