## [Peer review file · Nature Communications]

Reviewers' comments:

Reviewer #1 (Remarks to the Author):

The manuscript from Araya-Secchi investigates the structure of EC repeats 8-10 of protocadherin 15, one of the components of the tip link. Like the EC1-EC2 structure reported by the senior author, this construct of three EC domains is monomeric. More significantly, the structure shows a remarkable 90° bend, which could impart some elasticity to the tip link (albeit its estimated stiffness is greater than the gating spring, the elastic element that transmits force to the transduction channel). This structure is surprising and will be of substantial interest to not only auditory neuroscientists, but structural biologists as well. With some minor issues of interpretation, discussed below, the manuscript is convincing and does not require further evidence to strengthen its conclusions.

Major comments

1. The authors should discuss better the relationship between the kink they describe, which should be relatively close to the membrane insertion of PCDH15, with the rapid-freeze, deep-etch images presented by Kachar et al. (2000). Interestingly, many of those high-resolution images-the highest resolution views of the tip link-show a bulge not too far from the membrane, which is where you would expect to see the kink (e.g., Fig. 1B, 2A-C). On the other hand, the images of Fig. 3C-D of Kachar et al. show what is interpreted as branching near the membrane insertion, but might be a manifestation of kinking (especially Fig. 3D).

2. As noted above, neither EC1-2 nor EC8-10 show dimerization, which has been a hallmark of the tip link model since Kachar (2000). Could this model be wrong? Could the tip link be a heterodimer of one PCDH15 and one CDH23? That's a heretical idea but maybe needs to be discussed.

3. It needs to be emphasized that not only does the estimated stiffness of this construct seem way too high to be the gating spring, but also the stretching distance seems far too short. The data in Assad & Corey (1992) and especially Shepherd et al. (1994) indicate that the gating spring may be able to extend 100-200 nm, far more than the ~5 nm for the E9-E10 kink. It is true that there is "[an] elastic element in the protocadherin-15 tip link of the inner ear," (the manuscript title, of course) but it is not THE elastic element we have been searching for. I agree that the kink is very interesting, but the title is a bit misleading. I'm not sure that it needs to be changed, but I think it needs to be emphasized in the abstract as well as the text that the properties of the kink do not match with the properties of the gating spring.

4. I agree that the kink bend might account for the "release element" that has been discussed. However, one needs to come up with a way in which to make the kink release in a nonlinear way. The release element idea works (in theory at least) because increased force leads to opening of the channel and the increase in Ca²⁺-a state change for the channel, if you will-triggers the release. If the kink is linearly elastic, it cannot account for the release element. If it bends all at once, which doesn't look to be the case from the MD simulations but I'm not sure, then it would work. Please discuss more.

Reviewer #2 (Remarks to the Author):

Summary of the key results: The authors report the crystal structure of the EC8-EC10 fragment of protocadherin-15 at 2.8 Å resolution. While the EC8-EC9 linker region is canonical (i.e. calcium-binding) and therefore essentially rigid and linear, the EC9-EC10 linker region does not contain any calcium binding site and it is therefore calcium free, a feature that allows structural flexibility at the level of the EC9-EC10 tandem repeat and alters the typical linear arrangements of the

cadherin EC repeats. MD simulations and SAXS data are also provided by the authors, confirming the non-linear conformation of the PCDH15 EC8-EC10 fragment observed in the crystal.

Major comments:

1) As the sequence analysis of the EC9-EC10 linker indicates the absence of calcium-binding motifs, the flexibility of the calcium free EC9-EC10 linker region is actually not surprising. The bending is similar to that observed in a structure of *Drosophila* N-cadherin, which is also calcium free. The structure confirms that cadherin rigidity is provided by calcium ions and that lack of calcium coordination results in structural flexibility. Moreover, the buried surface area between EC9 and EC10 is quite small (389 Å²) and the contacts that stabilize the observed conformation are mostly hydrophobic in nature, suggesting that they are essentially non-specific. Therefore, beyond the flexibility of the EC9-EC10 hinge region, this crystal structure does not appear to provide any mechanistic information on inner ear mechanotransduction.

2) Beside comparing the structure shown here with the structure of *Drosophila* N-cadherin mentioned above, it would also be informative to compare it with the crystal structure of calcium free T-cadherin EC1EC2 (Ciatto, Nat.Struct.Mol.Biol. 2010). This flexible T-cadherin fragment was crystallized after removing the calcium ions from a canonical calcium binding linker region. Although the structure does not appear to be deposited in the PDB (or at least I have not been able to find it), it would be interesting to discuss it in the context of the novelty of the claims that the author of the present paper make regarding the PCDH15 EC8-EC10 fragment.

3) The authors do not provide any discussion on the crystal packing arrangement of the EC8-EC10 fragment and on the intermolecular contacts that this fragment forms within the crystal lattice. I think a detailed discussion on the quaternary structure of this 3-domain fragment would be interesting, even more so when compared to the packing arrangement of the EC9-EC10 structure alone, for which the author show no data. The authors state that they solved the structure of this 2-domain portion as well and claim that the relative orientation of EC9 and EC10 are the same in the two structures. However, a comparison between the packing arrangement of the two structures would be informative and would probably show whether different intermolecular contacts are formed in the two crystal lattices. This information would possibly provide conclusive experimental evidence of the effects (or lack of effects) of crystal forces on the molecular conformation.

I believe that the manuscript lacks the level of novelty required by Nature Communications, especially considering the little, if any, mechanistic information provided. The paper seems to be suitable for a more specialized journal.

Reviewer #3 (Remarks to the Author):

This manuscript resolves the structure and force response of a three domain segment (EC8-10) of protocadherin-15 (PCDH15), a key component of tip links that mediate perception of sound and head movements. Using X-ray crystallography, the authors show that the EC8-10 domains are arranged non-linearly. They confirm this non-linear orientation using force-free Molecular Dynamics (MD) simulations and small angle X-ray scattering experiments. Finally using steered MD simulations, they show that the unbending of EC9-10 confers elasticity to tip links. This manuscript provides the first view of PCDH15's non-canonical EC linkers and strongly suggests that the bent, calcium-free linker in EC9-10 plays a mechanical role in hair-cell transduction. The range of techniques employed in this investigation are very impressive, the analysis is very thorough and the biological importance of the findings are high. Consequently, I believe this manuscript will be of much interest to the readership of Nature Communications. However, I have several suggestions

that I think will make the results more convincing.

1) My most significant comment involves the SMD simulations that were used to study the forced unbending and unfolding of the EC8-10 domains. In these simulations, a constant-force was applied to the N-terminus (EC8) and the C-terminus (EC10) as shown in Figure 6d. Given the bent conformation adopted by EC9-10 linker, this pulling force essentially exerts a torque on the linker and unbends it. The simulations show a complete unbending of EC9-10 at 25 pN, 50 pN and 100 pN while no unbending was measured at 10 pN suggesting that this bent conformation exists in native tip links that experience a resting tension of ~ 10 pN. Based on this result, the authors conclude that the bent section of the protein may serve as part of the hair cell "gating-spring". However what the authors do not consider is that in the tip-link, the PCDH15 lever arm is longer with at least one (EC11) and maybe even two additional EC domains. Since the applied torque depends on the length of the lever arm, it is very conceivable that due to the presence of additional EC domains in the lever arm, a 10 pN force would likely unbend the EC9-10 linker. The authors need to directly address this issue. One way of doing this would be to mimic the 'real' lever arm by docking either 1 or 2 additional EC domains at the C-terminal end of EC9-10 and then performing the SMD simulations. This should also be repeated for the constant velocity stretching simulations shown in Figure 7.

2) The chimera that the authors use to measure the forced unbending and unbinding of PCDH15 with CDH23 is rather simplistic. This chimera was generated by coupling the PCDH15 EC1-2 domains to EC8-10 via an artificial canonical linker. I have some reservations on how faithfully this 'hypothetical' structure represents the coupling between the unbending/unfolding of the EC9-10 and unbinding of the tetrameric CDH23-PCDH15 complex. What steps did the authors take to ensure that the orientation of EC8 with respect to EC2 is the same as the orientation of EC2-3? This could affect the force dependent unbending and unbinding of PCDH15. Furthermore, the linker between the EC5-6 domains is also calcium free. The simulations ignore likely effect of the calcium free EC5-6 linker on the overall unbending-unbinding mechanism.

3) The authors measure a twist between EC8 and EC9 which agrees with TEM images showing that neighboring PCDH15s intertwine helically to form a dimer (Reference 27). How does the twisting of EC8-9 within a dimer affect the unbending of EC9-10 linker region?

4) Figure 2E showing the twist of EC9 relative to EC8 is very hard to understand. Although the bending of EC9 relative to EC8 is easily visualized in the upper left panel of 2E, the rotation of EC9 in the x-y plane is not as clear. It will be better if the twisting is represented in three-dimensions (similar to the upper left panel of 2E) rather than in two-dimensions.

5) The authors claim that their MD simulations show a rather stable bent structure and identify only local deformation of supporting loop residues (1040-1046). However the RMSFs in Supplementary figure 5b show considerable motion of the entire EC10 domain, not just the supporting loops. Can the authors clarify this? The authors should also show the RMSDs of the 3-10 helix and EC10 support loop in either figure 4 or the SI.

6) In the long timescale relaxation of the EC9-10 stretched states (shown in Supplementary figure 6b), there are very large step-like transitions in the end-to-end angles. Some of these transitions seem unphysical and involve angular changes of >100 degrees occurring over a few frames of the simulations. Can the authors explain these angular changes?

7) In the constant velocity SMD simulations that show the forced unbending and unfolding of EC8-10, I would have expected to observe a large increase in the end-to-end distance at low force as the EC9-10 linker straightened. This is not obvious in Figure 7a. Perhaps the authors should include an inset in this figure showing this feature. Furthermore, in figure 7a, the authors need to indicate the regions that correspond to phase 1 and phase 2.

8) I am confused by the method used to measure the spring constant of EC8-10 (i.e. by measuring the slope of the force vs. end-to-end distance). Since a polypeptide chain is not a Hookean spring but rather a non-linear spring, why does the slope correspond to the stiffness of EC8-10? In fact, the stiffness measured at the two lowest velocities vary significantly. Why would the stiffness vary with pulling velocity?

9) On page 7, line 17, the authors state that "While the buried surface area of this interface is small, a second structure of PCDH15 EC9-10 (without EC8) shows the same bent conformation (data not shown), suggesting that the EC9-10 interface is robust." The authors need to show this data or eliminate this statement.

10) The simulations with the chimeric protein shows that the EC9-10 linker does not unfold. In figure 8c, the constant velocity stretching at 1 nm/ns appears to show that unbinding occurs at a force of ~ 600 pN. However, the forced unfolding simulations in figure 7a, at an identical pulling velocity, shows that the protein unfolds at approximately the same force. How many times were each of these simulations repeated? Perhaps this is an artifact of poor statistics. However the authors should discuss this discrepancy.

11) Since the linker between the EC5-6 domains is also calcium free, it likely has an effect on the overall elasticity of the tip links. Since the EC6-6 and EC9-10 linkers are in series, the softest linker will dominate tip link elasticity. The authors should include a discussion of this in the manuscript.

12) Since the EC8-9 calcium binding linker generates a twisting of these domains already implies non-linear arrangement of the whole 11 domains. Therefore it is not correct to say two interacting PCDH15 are oriented in a "linear" or "parallel" orientation.

13) On page 7, the authors state that most of the residues in the hydrophobic core and linker are highly conserved. However a careful examination of figure 3c shows that several residues, most notably E1092, E1010 and I092 are not very well conserved at all. The authors should be more precise in their claims.

14) In page 18 (line 23-24) of the discussion section, please provide a reference to the statement that the EC5-6 linker lacks calcium-binding residues and might be flexible or adopt a bent conformation as well.

RESPONSE

We thank all three reviewers for their thorough and thoughtful comments. Following their suggestions, we have finalized refinement and deposited a new structure of mouse Pcdh15 EC9-10 (PDB code 5KJ4), which strengthens our previous findings and is now discussed in the revised manuscript. We have also performed one additional equilibrium simulation (500 ns), we have limited the analysis of the other one to avoid biases from periodic image contacts (occurring at $t > 400$ ns), and we have added seven new SMD simulations to increase the number of replicates and ensure that simulation predictions are robust. These new data have been incorporated to our manuscript along with edits (marked blue) that help clarify and address all reviewer's requests and concerns.

A detailed response follows, including reviewer comments in *italics*.

Reviewer #1 (Remarks to the Author):

The manuscript from Araya-Secchi investigates the structure of EC repeats 8-10 of protocadherin 15, one of the components of the tip link. Like the EC1-EC2 structure reported by the senior author, this construct of three EC domains is monomeric. More significantly, the structure shows a remarkable 90° bend, which could impart some elasticity to the tip link (albeit its estimated stiffness is greater than the gating spring, the elastic element that transmits force to the transduction channel). This structure is surprising and will be of substantial interest to not only auditory neuroscientists, but structural biologists as well. With some minor issues of interpretation, discussed below, the manuscript is convincing and does not require further evidence to strengthen its conclusions.

We share the reviewer's enthusiasm about our findings and have revised the manuscript to address the issues of interpretation raised in the reviewer comments below.

Major comments

1. The authors should discuss better the relationship between the kink they describe, which should be relatively close to the membrane insertion of PCDH15, with the rapid-freeze, deep-etch images presented by Kachar et al. (2000). Interestingly, many of those high-resolution images-the highest resolution views of the tip link-show a bulge not too far from the membrane, which is where you would expect to see the kink (e.g., Fig. 1B, 2A-C). On the other hand, the images of Fig. 3C-D of Kachar et al. show what is interpreted as branching near the membrane insertion, but might be a manifestation of kinking (especially Fig. 3D).

This is an interesting observation that is now briefly included in the main text:

“Interestingly, the highest resolution views of tip links *in situ* show what could be interpreted as a bent EC9-10 linker near the point of membrane insertion of PCDH15 (Figure 3D in⁶).” (page 13, line 23)

2. As noted above, neither EC1-2 nor EC8-10 show dimerization, which has been a hallmark of the tip link model since Kachar (2000). Could this model be wrong? Could the tip link be a heterodimer of one PCDH15 and one CDH23? That's a heretical idea but maybe needs to be discussed.

The reviewer is right in that both EC1-2 and EC8-10 are monomeric in solution. This may suggest that the tip link is made of only one PCDH15 and one CDH23 or, more likely, that dimerization is mediated by other EC repeats. We have included a brief comment discussing this issue along with our discussion of crystallographic contacts in the supplement.

“Parallel dimerization of PCDH15 and CDH23 is supported by biochemical data obtained using full-length extracellular domains¹, yet CDH23 EC1-2^{2,3}, PCDH15 EC8-10, and mmPcdh15 EC9-10 are monomeric in solution. Crystallographic contacts for the EC8-10 and EC9-10 structures seem to support the monomeric states of these fragments. ... [P]arallel dimerization of PCDH15 might be mediated by other EC repeats, or by multiple small non-specific contacts as observed by negative staining transmission electron microscopy¹.” (Supplementary Material, page 2, line 5)

3. It needs to be emphasized that not only does the estimated stiffness of this construct seem way too high to be the gating spring, but also the stretching distance seems far too short. The data in Assad & Corey (1992) and especially Shepherd et al. (1994) indicate that the gating spring may be able to extend 100-200 nm, far more than the ~5 nm for the E9-E10 kink. It is true that there is "[an] elastic element in the protocadherin-15 tip link of the inner ear," (the manuscript title, of course) but it is not THE elastic element we have been searching for. I agree that the kink is very interesting, but the title is a bit misleading. I'm not sure that it needs to be changed, but I think it needs to be emphasized in the abstract as well as the text that the properties of the kink do not match with the properties of the gating spring.

We agree with the reviewer and have modified the abstract, within the space constraints, to avoid misleading the reader. The new abstract now says:

“Simulations also suggest that unbending of EC9-10 confers some elasticity to otherwise rigid tip links.” (page 1, line 18)

In addition, we have modified the discussion section to include:

“Electrophysiological measurements suggest that if the tip link is the gating spring, it might stretch by 200 nm for extreme mechanical deflection of hair bundles⁵⁰. Unbending only provides a 5 nm extension, implying that such large stretching would either require unfolding of EC repeats or membrane tether formation at the tip link insertion point^{51,52}. Regardless of the mechanism, the elastic response of the whole transduction apparatus will be dominated by the softest component, yet to be determined. Clearly, more complete structural models of the hair cell transduction apparatus are required to dissect out the exact contribution of each of its components to the gating spring elasticity, with unbending TSE of PCDH15 EC9-10 being part of it.” (page 14, line 27)

We also note that the values for the effective spring constant of EC9-10 predicted from simulations represent an upper boundary, as slower pulling simulations (not achievable with current supercomputers) may reveal a softer response, as now discussed in the text:

“[T]he spring constant estimates from simulations represent an upper boundary to physiological values, as stretching speeds *in silico* are significantly faster than those used in the experiments that were used to estimate the gating spring constant (see Supplementary Discussion).” (page 14, line 20)

4. I agree that the kink bend might account for the "release element" that has been discussed. However, one needs to come up with a way in which to make the kink release in a nonlinear way.

The release element idea works (in theory at least) because increased force leads to opening of the channel and the increase in Ca²⁺-a state change for the channel, if you will-triggers the release. If the kink is linearly elastic, it cannot account for the release element. If it bends all at once, which doesn't look to be the case from the MD simulations but I'm not sure, then it would work. Please discuss more.

In the original model of Ca²⁺-dependent channel reclosure (Appendix of Martin et al., 2003), the release element is incorporated as a linear spring, with a spring constant κ_{RE} that changes depending on calcium concentration:

$$\kappa_{RE} = (1 - p_{B,RE})(\kappa_{RE,MAX} - \kappa_{RE,MIN}) + \kappa_{RE,MIN},$$

where $p_{B,RE}$ is the binding probability of calcium to the release element, and the stiffness of the reclosure element varies linearly between a maximal and a minimal value ($\kappa_{RE,MAX}$ and $\kappa_{RE,MIN}$). The non-linearity arises from the calcium-dependency of the stiffness. In our manuscript we speculate that calcium binding to other parts of the tip link could trigger unbending, thereby changing the tip link elasticity. This is not incompatible with the linear response for EC8-10 observed in simulations, but remains a speculation and the text has been modified to clarify how unbending could account for the release element:

“In this speculative scenario, PCDH15 EC9-10 would play a role similar to that of a hypothetical “release element” thought to mediate fast, calcium-dependent adaptation^{53,54}. Calcium would have to regulate this process allosterically through other PCDH15 and CDH23 EC repeats with calcium-binding motifs, thereby introducing a non-linear response and modulating the stiffness of the release element as proposed by Martin et al⁵⁴.” (page 15, line 11)

Reviewer #2 (Remarks to the Author):

Summary of the key results: The authors report the crystal structure of the EC8-EC10 fragment of protocadherin-15 at 2.8 Å resolution. While the EC8-EC9 linker region is canonical (i.e. calcium-binding) and therefore essentially rigid and linear, the EC9-EC10 linker region does not contain any calcium binding site and it is therefore calcium free, a feature that allows structural flexibility at the level of the EC9-EC10 tandem repeat and alters the typical linear arrangements of the cadherin EC repeats. MD simulations and SAXS data are also provided by the authors, confirming the non-linear conformation of the PCDH15 EC8-EC10 fragment observed in the crystal.

Major comments:

1) As the sequence analysis of the EC9-EC10 linker indicates the absence of calcium-binding motifs, the flexibility of the calcium free EC9-EC10 linker region is actually not surprising. The bending is similar to that observed in a structure of Drosophila N-cadherin, which is also calcium free. The structure confirms that cadherin rigidity is provided by calcium ions and that lack of calcium coordination results in structural flexibility. Moreover, the buried surface area between EC9 and EC10 is quite small (389 Å²) and the contacts that stabilize the observed conformation are mostly hydrophobic in nature, suggesting that they are essentially non-specific. Therefore, beyond the flexibility of the EC9-EC10 hinge region, this crystal structure does not appear to provide any mechanistic information on inner ear mechanotransduction.

The reviewer is right in that the lack of calcium-binding motifs in the sequence of PCDH15 is enough to predict some type of bending as reported in Ciatto et al. (2010), Jin et al. (2012), Tsukasaki et al (2014), and Tariq et al. (2015). However, bending is different for PCDH15 EC9-

10 (~90°, L-shape), DNa cad EC2-3 (~80°, V-shape), and CDH13 (Tcad) in calcium-free conditions (U-shape). An additional panel in Supplementary Figure 6 illustrates this point (Supplementary Figure 6a-d). To date no common pattern has been found among calcium-free inter-repeat linkers that allow for the prediction (from sequence alone) of the degree of bending and flexibility of the structures carrying them. Without our structures and simulations of EC9-10 it would be difficult to predict the ~90° bending, the mechanical properties of these repeats, the interfacial residues, and the existence of a 3_{10} helix at the linker. These differences have been now highlighted in the text:

“The bending at EC9-10 is similar to that observed in a structure of *Drosophila* N-cadherin (DNa cad), which also has a calcium-free linker (PDB codes 3UBH, 3UBG)³⁵, and to the conformation adopted by T-cadherin (Tcad/CDH13) in calcium-free conditions (3KR5)⁴⁰. However, the PCDH15 EC9-10 linker is L-shaped, while DNa cad is V-shaped, and Tcad without calcium is U-shaped (Supplementary Fig. 6a-d). In addition, the PCDH15 EC9-10 linker has structural features not observed in DNa cad, Tcad, or any other cadherin structures⁴¹⁻⁴³. A unique EC9-10 3_{10} helix is located in the middle of the EC9-10 linker (His 1007 – Ile 1011; blue arrow in Fig. 1d&e), and an atypical EC10 FG- α loop (Leu 1091 – Asn 1105; red arrow in Fig. 1d&e) form an EC9-10 interface that stabilizes the observed bent conformation of the linker.” (page 5, line 10)

2) Beside comparing the structure shown here with the structure of Drosophila N-cadherin mentioned above, it would also be informative to compare it with the crystal structure of calcium free T-cadherin EC1EC2 (Ciatto, Nat.Struct.Mol.Biol. 2010). This flexible T-cadherin fragment was crystallized after removing the calcium ions from a canonical calcium binding linker region. Although the structure does not appear to be deposited in the PDB (or at least I have not been able to find it), it would be interesting to discuss it in the context of the novelty of the claims that the author of the present paper make regarding the PCDH15 EC8-EC10 fragment.

Following the reviewer suggestion, we compared EC9-10 from our structure with the calcium-free T-cadherin EC1-2 structure (Ciatto et al., 2010; PDB code: 3K5R). This comparison (modified Supplementary Fig. 6) highlights the uniqueness of our structure and the novelty it presents to the structural diversity of cadherins. The text has been modified to discuss this as indicated in our response to the first major comment of this reviewer (see also page 5, line 10).

3) The authors do not provide any discussion on the crystal packing arrangement of the EC8-EC10 fragment and on the intermolecular contacts that this fragment forms within the crystal lattice. I think a detailed discussion on the quaternary structure of this 3-domain fragment would be interesting, even more so when compared to the packing arrangement of the EC9-EC10 structure alone, for which the author show no data. The authors state that they solved the structure of this 2-domain portion as well and claim that the relative orientation of EC9 and EC10 are the same in the two structures. However, a comparison between the packing arrangements of the two structures would be informative and would probably show whether different intermolecular contacts are formed in the two crystal lattices. This information would possibly provide conclusive experimental evidence of the effects (or lack of effects) of crystal forces on the molecular conformation.

We have finalized and deposited a structure for mouse Protocadherin-15 EC9-EC10 (mmPcdh15 EC9-10, 5KJ4). An analysis of this new structure and the packing arrangement for both human PCDH15 EC8-10 and mmPcdh15 EC9-10 is now included in the text (page 6, line 1), the supplementary text and figures (Supplementary Material, page 2, line 5; Supplementary Fig. 7).

I believe that the manuscript lacks the level of novelty required by Nature Communications, especially considering the little, if any, mechanistic information provided. The paper seems to be suitable for a more specialized journal.

We hope that this revised version addressing all reviewers comments highlights the unique mechanistic information provided by our work, including the EC9-10 L-shaped kink, the main residues involved in the EC9-EC10 interface, and the predicted elasticity of this fragment, which is relevant for hair-cell mechanotransduction. Given the abundance of calcium-free EC repeats in the cadherin family and the relevance of PCDH15 in hearing^{26,27}, Usher syndrome (including blindness), cancer (Rouget-Quermalet et al., *Oncogene* 2006), and other diseases (Huertas-Vazquez et al., *Hum. Genet.* 2010), we believe that a non-specialized journal, such as *Nature Communications*, is appropriate.

Reviewer #3 (Remarks to the Author):

This manuscript resolves the structure and force response of a three domain segment (EC8-10) of protocadherin-15 (PCDH15), a key component of tip links that mediate perception of sound and head movements. Using X-ray crystallography, the authors show that the EC8-10 domains are arranged non-linearly. They confirm this non-linear orientation using force-free Molecular Dynamics (MD) simulations and small angle X-ray scattering experiments. Finally using steered MD simulations, they show that the unbending of EC9-10 confers elasticity to tip links. This manuscript provides the first view of PCDH15's non-canonical EC linkers and strongly suggests that the bent, calcium-free linker in EC9-10 plays a mechanical role in hair-cell transduction. The range of techniques employed in this investigation are very impressive, the analysis is very thorough and the biological importance of the findings are high. Consequently, I believe this manuscript will be of much interest to the readership of Nature Communications. However, I have several suggestions that I think will make the results more convincing.

We thank the reviewer for their comments and positive appraisal of our work and for the thorough review and critique. We have incorporated all suggestions and hope that the revised version is satisfactory.

1) My most significant comment involves the SMD simulations that were used to study the forced unbending and unfolding of the EC8-10 domains. In these simulations, a constant-force was applied to the N-terminus (EC8) and the C-terminus (EC10) as shown in Figure 6d. Given the bent conformation adopted by EC9-10 linker, this pulling force essentially exerts a torque on the linker and unbends it. The simulations show a complete unbending of EC9-10 at 25 pN, 50 pN and 100 pN while no unbending was measured at 10 pN suggesting that this bent conformation exists in native tip links that experience a resting tension of ~10 pN. Based on this result, the authors conclude that the bent section of the protein may serve as part of the hair cell "gating-spring". However what the authors do not consider is that in the tip-link, the PCDH15 lever arm is longer with at least one (EC11) and maybe even two additional EC domains. Since the applied torque depends on the length of the lever arm, it is very conceivable that due to the presence of additional EC domains in the lever arm, a 10 pN force would likely unbend the EC9-10 linker. The authors need to directly address this issue. One way of doing this would be to mimic the 'real' lever arm by docking either 1 or 2 additional EC domains at the C-terminal end of EC9-10 and then performing the SMD simulations. This should also be repeated for the constant velocity stretching simulations shown in Figure 7.

This is an excellent point that we had not discussed in our manuscript. Unfortunately, it is unclear that the PCDH15 lever arm will be longer, even with EC11, as the human EC10-11 linker lacks one of the residues involved in calcium binding (DXNDN is replaced by DXNNH), which may affect bending and flexibility. We have included a brief comment about this point in the discussion:

“We note that EC11 and a straight EC10-EC11 linker would provide a longer “lever arm” that would result in a larger applied torque at the same force, which may favor unbending at smaller forces. However, the human EC10-11 linker lacks one of the residues involved in calcium binding (Supplementary Fig. 2), and might have enhanced flexibility that would hamper the action of a longer lever arm (Supplementary Fig. 15).” (page 14, line 5)

We are working on obtaining further structures that could address this point and consider that the new discussion text is sufficient to alert the reader about the possible effects of additional EC repeats.

2) The chimera that the authors use to measure the forced unbending and unbinding of PCDH15 with CDH23 is rather simplistic. This chimera was generated by coupling the PCDH15 EC1-2 domains to EC8-10 via an artificial canonical linker. I have some reservations on how faithfully this 'hypothetical' structure represents the coupling between the unbending/unfolding of the EC9-10 and unbinding of the tetrameric CDH23-PCDH15 complex. What steps did the authors take to ensure that the orientation of EC8 with respect to EC2 is the same as the orientation of EC2-3? This could affect the force dependent unbending and unbinding of PCDH15. Furthermore, the linker between the EC5-6 domains is also calcium free. The simulations ignore likely effect of the calcium free EC5-6 linker on the overall unbending-unbinding mechanism.

The chimeric structure built and presented in our manuscript joins together all the currently published structures of PCDH15 and CDH23. The reviewer is right in that the chimeric link is artificial and the orientation of EC8 with respect to EC2 is not the same as the orientation of EC2-3. The chimeric model is not aimed at representing an EC2-3 linker. This chimeric model is used to illustrate the behavior of EC8-10 and does not take into account the contributions of other EC repeats and unusual linkers, as explicitly discussed now in the text:

“Simulations of an artificial chimeric model involving all published structures of tip link EC repeats suggest that unbending of EC9-10 is followed by unbinding without any relevant unfolding, as observed for other classical cadherins^{56,57}. However, this artificial model does not take into account the effect of parallel dimerization (see below) and does not incorporate the behavior of other EC repeats and atypical linkers that may unfold before unbinding occurs.” (page 15, line 22)

Otherwise, this artificial linker faithfully represents a canonical linker, as it contains all the required calcium-binding motifs, it features the correct coordination of its 3 calcium ions (as checked by “check-my-metal”⁷⁴), and it allowed us to compare the forces necessary to unbind the complex, to unbend the EC9-10 interfaces, and to unfold EC10. As more structures are solved, we will build and compare these results with more complete models of the entire PCDH15 extracellular domain.

3) The authors measure a twist between EC8 and EC9 which agrees with TEM images showing that neighboring PCDH15s intertwine helically to form a dimer (Reference 27). How does the twisting of EC8-9 within a dimer affect the unbending of EC9-10 linker region?

This is a very interesting question, but we do not have a structure or a model of the parallel dimer formed by PCDH15 yet. Our crystal structures of EC8-10 and EC9-10 do not show dimerization in solution (see response to reviewer #1 point #2 above). Therefore, we do not know how the twisting and untwisting between EC8-9 will affect the unbending of EC9-10 in a dimer complex.

4) Figure 2E showing the twist of EC9 relative to EC8 is very hard to understand. Although the bending of EC9 relative to EC8 is easily visualized in the upper left panel of 2E, the rotation of EC9 in the x-y plane is not as clear. It will be better if the twisting is represented in three-dimensions (similar to the upper left panel of 2E) rather than in two-dimensions.

Illustration of the tilt and twist of EC repeats is difficult, and we have included three-dimensional representations in the supplement where a larger format can be used (Supplementary Fig. 5).

5) The authors claim that their MD simulations show a rather stable bent structure and identify only local deformation of supporting loop residues (1040-1046). However the RMSFs in Supplementary figure 5b show considerable motion of the entire EC10 domain, not just the supporting loops. Can the authors clarify this? The authors should also show the RMSDs of the 3-10 helix and EC10 support loop in either figure 4 or the SI.

The RMSFs in Supplementary Figure 5b (now Supplementary Figs. 8b and 9b) show considerable motion of the entire EC10 only when the entire EC8-10 structure is used as a reference for the structural alignment needed for the computation of the fluctuations. This is an artifact of the poor alignment resulting from inter-repeat motions. A more accurate view of the fluctuations in each repeat is represented by the colored lines (red, green, and blue for EC8, EC9, and EC10 respectively), which were obtained using EC-based structural alignments. The colored curves show similar fluctuations across all EC repeats. This has been clarified in the caption of Supplementary Figures 8 and 9.

We have also included two new panels (Supplementary Figs. 8c,d and 9c,d) with the requested data.

6) In the long timescale relaxation of the EC9-10 stretched states (shown in Supplementary figure 6b), there are very large step-like transitions in the end-to-end angles. Some of these transitions seem unphysical and involve angular changes of >100 degrees occurring over a few frames of the simulations. Can the authors explain these angular changes?

The step-like transitions mentioned by the reviewer are not unphysical and occur over hundreds of frames of simulation (saved every 240 picoseconds) representing tens of nanoseconds with a time step of 2 femtoseconds. The apparent step-like changes result from the scale used in the plot that accommodates the entire length of the simulation. We have added a third panel in Supplementary Fig. 10 (panel c) that corresponds to a detail of the plot presented in panel b, illustrating this point.

7) In the constant velocity SMD simulations that show the forced unbending and unfolding of EC8-10, I would have expected to observe a large increase in the end-to-end distance at low force as the EC9-10 linker straightened. This is not obvious in Figure 7a. Perhaps the authors should include an inset in this figure showing this feature. Furthermore, in figure 7a, the authors need to indicate the regions that correspond to phase 1 and phase 2.

A new supplementary figure (Supplementary Fig. 12) has been added. This figure illustrates the

change in end-to-end distance for each phase as requested. Phases I and II have been indicated in Fig. 8c, and Supplementary Figs. 12 and 14b.

8) I am confused by the method used to measure the spring constant of EC8-10 (i.e. by measuring the slope of the force vs. end-to-end distance). Since a polypeptide chain is not a Hookean spring but rather a non-linear spring, why does the slope correspond to the stiffness of EC8-10? In fact, the stiffness measured at the two lowest velocities vary significantly. Why would the stiffness vary with pulling velocity?

A polypeptide chain is not a Hookean spring, but we can estimate an “effective Hookean stiffness” from the linear regions of the force vs. distance plots. This has been clarified in page 11, line 27. The aforementioned differences in stiffness arise due to viscous drag at different speeds, and the dynamic response of bonds under various loading rates. This is a well-known effect that has been studied theoretically and experimentally (Evans and Ritchie, 1997; Evans and Calderwood, 2007) and that also manifests itself in the rupture of bonds during unbending of EC9-10. This is explained in the Supplementary Discussion (Supplementary Material, page 3, line 29, references 16 to 19).

9) On page 7, line 17, the authors state that "While the buried surface area of this interface is small, a second structure of PCDH15 EC9-10 (without EC8) shows the same bent conformation (data not shown), suggesting that the EC9-10 interface is robust." The authors need to show this data or eliminate this statement.

The structure of mouse Pcdh15 EC9-10 has been deposited (PDB code 5KJ4). The revised manuscript includes figures and analyses of this new structure as requested (page 3, line 18; Table 1 and Supplementary Fig. 7).

10) The simulations with the chimeric protein shows that the EC9-10 linker does not unfold. In figure 8c, the constant velocity stretching at 1 nm/ns appears to show that unbinding occurs at a force of ~ 600 pN. However, the forced unfolding simulations in figure 7a, at an identical pulling velocity, shows that the protein unfolds at approximately the same force. How many times were each of these simulations repeated? Perhaps this is an artifact of poor statistics. However the authors should discuss this discrepancy.

In the original manuscript simulations at a given speed were not repeated. We have now performed independent repetitions of the simulations at 1 nm/ns for PCDH15 EC8-10 ($n = 4$) and the chimeric complex ($n = 3$) to obtain better statistics. The average force peak that marks the beginning of EC10 unfolding is 794 ± 108 pN for PCDH15 EC8-10, and the average force peak for unbinding in the chimeric complex is 652 ± 42 pN. These results show that the average force required to unbind the complex at this speed is lower than the force needed to unfold EC10. As stated in the manuscript, this trend was observed also for the SMD simulations at 0.1 nm/ns. We have added a supplementary figure (Supplementary Fig. 12) to show the variability across repetitions. Supplementary Fig. 14f has been updated with the standard deviations for these new runs.

11) Since the linker between the EC5-6 domains is also calcium free, it likely has an effect on the overall elasticity of the tip links. Since the EC6-6 and EC9-10 linkers are in series, the softest linker will dominate tip link elasticity. The authors should include a discussion of this in the manuscript.

The revised manuscript includes a statement that indicates the possibility of unbending at EC5-6:

“In PCDH15, the EC5-6 linker lacks calcium-binding residues and might unfold, be flexible, or adopt a bent conformation as well^{35,40,46,47}.” (page 14, line 25)

A sentence indicating that the softest linker will dominate tip-link elasticity has been included as well:

“Regardless of the mechanism, the elastic response of the whole transduction apparatus will be dominated by the softest component, yet to be determined.” (page 15, line 4)

12) Since the EC8-9 calcium binding linker generates a twisting of these domains already implies non-linear arrangement of the whole 11 domains. Therefore it is not correct to say two interacting PCDH15 are oriented in a "linear" or "parallel" orientation.

We have defined “parallel” in biochemical terms, with two parallel proteins having their N-termini pointing in the same direction, as now explained in the text:

“(“parallel” is defined here to describe two proteins with their N- to C-termini directions aligned).” (page 16, line 6)

13) On page 7, the authors state that most of the residues in the hydrophobic core and linker are highly conserved. However a careful examination of figure 3c shows that several residues, most notably E1092, E1010 and I092 are not very well conserved at all. The authors should be more precise in their claims.

To address this issue we have included the following text:

“Conservation analysis of the interface reveals that most of the hydrophobic residues are highly conserved, while some of the polar and charged residues present in the EC9-10 linker and in the EC10 FG- α loop show more variability (Fig. 3c).” (page 5, line 25)

14) In page 18 (line 23-24) of the discussion section, please provide a reference to the statement that the EC5-6 linker lacks calcium-binding residues and might be flexible or adopt a bent conformation as well.

We have included references to the work of Jin et al.³⁵, Ciatto et al.⁴⁰, and Tariq et al.⁴⁶ where analyses of flexible linkers are discussed, and a reference to Tsukasaki et al.⁴⁷, where an analysis of the pcdh15 linkers is presented in the supplement. This is discussed in page 14, line 25.

REVIEWERS' COMMENTS:

Reviewer #1 (Remarks to the Author):

The authors have satisfactorily addressed my comments.

Reviewer #2 (Remarks to the Author):

I am satisfied with the changes that the authors have made to the previous version of the manuscript and I recommend the paper for publication in the present form.

Reviewer #3 (Remarks to the Author):

The authors have been very thorough in revising the manuscript based on the reviewers feedback. I am satisfied with the revisions and recommend that the revised manuscript be published in Nature Communications.

RESPONSE

We thank all three reviewers for their time and final approval of our submission. Since there were no issues raised by reviewers (see below), we have focused our edits on editorial requests to comply with *Nature Communications* Policies and Format (see letter to editor for details).

REVIEWERS' COMMENTS:

Reviewer #1 (Remarks to the Author): The authors have satisfactorily addressed my comments.

Reviewer #2 (Remarks to the Author): I am satisfied with the changes that the authors have made to the previous version of the manuscript and I recommend the paper for publication in the present form.

Reviewer #3 (Remarks to the Author): The authors have been very thorough in revising the manuscript based on the reviewers feedback. I am satisfied with the revisions and recommend that the revised manuscript be published in Nature Communications.